# ALYREF condensation stabilizes m5C-modified PARP10 mRNA and promotes PI3K-AKT signaling in ovarian cancer

Hongyan Zhao[1,2,3,9], Qinglv Wei [4,9], Zhi Luo[5,9], Xiaoyi Liu[1,9], Chenyue Yang[1], Ningxuan Chen[3], Yuan Wang[3], Xin Luo[1], Xinzhao Zuo[3], Qingya Luo[6], Yu Yang[1], Yang Zhou[2], Jiaqi Liu[2], Te Zhang[2], Dan Yang[1], Yingfei Long[7], Youchaou Mobet[8], Jing Xu[3], Wei Wang [5✉], Tao Liu [1,3✉] & Ping Yi [1,3✉]

## Abstract

**The role of epigenetic regulation of RNAs in the tumorigenesis remains incompletely understood. This study uncovers a critical function of the 5-methylcytosine (m5C) RNA modification reader protein ALYREF (also termed, ALY; BEF) in ovarian cancer. ALYREF is elevated in ovarian cancer patient samples, and its depletion reduces ovarian tumorigenesis and metastasis in mice in a m5C-dependent manner. Mechanistically, ALYREF binds to the m5C-modified mRNA of ADP-ribosyltransferase PARP10, competing with exosome complex component MTR4, and enhancing the stability and nuclear export of PARP10 mRNA. Further, ALYREF forms condensates in the nucleus of ovarian cancer cells, and depletion or mutation of ALYREF's intrinsically disordered regions rescues its control on PARP10 mRNA nucleoplasmic distribution and stability, reduces tumor growth and is required for promotion of ovarian cancer aggressiveness and proliferation. Finally, ALYREF and PARP10 expression correlate with poor prognosis in ovarian cancer patients. Together, these findings suggest that ALYREF phase separation facilitates the malignant progression of ovarian cancer by promoting PARP10 expression and thereby enhancing PARP10-dependent proliferative pathways in a m5C-dependent manner.**

**Keywords** ALYREF; Phase Separation; m5C; RNA Stability
**Subject Categories** Cancer; Chromatin, Transcription & Genomics; RNA Biology

## Introduction

In eukaryotes, the expression of protein-coding genes is regulated by mRNA transcription, nuclear export, and degradation (Tudek et al, 2019). Eukaryotes have evolved sophisticated mechanisms of coordinating the nuclear export of RNAs, which is critical for gene expression (Fan et al, 2018; Hautbergue, 2017; Heath et al, 2016). The transcription-export (TREX) complex is an evolutionarily conserved multiprotein complex that plays a key role in mRNA biogenesis, including mRNA nuclear export, decay, transcription, and processing (Katahira, 2012). The RNA binding function of the TREX multi-subunit complex is performed by ALYREF (RNA and export factor binding proteins, REF; also known as ALY), which functions in exporting mRNAs from the nucleus and maintaining RNA stability (Chi et al, 2013; Fan et al, 2017). ALYREF also functions as a 5-methylcytosine (m5C) "reader" that recognizes m5C-modified RNA and regulates RNA nuclear export to the cytoplasm. m5C modification is a dynamic and reversible modification that is widely present in mRNA, tRNA, and rRNA (Bohnsack et al, 2019; David et al, 2017; Winans and Beemon, 2019). TRM4B and NSUN2 are m5C mRNA methylation transferases, and Y-box binding protein (YBX1) recognizes m5C-modified mRNAs and regulates the fate of RNA (Chen et al, 2019; David et al, 2017; Yang et al, 2017; Yang et al, 2019; Zou et al, 2020).

Biomolecular condensates are executors of spatial regulation and coordination of cell biological activities. Specific molecules in condensates maintain the liquid state by liquid-liquid phase separation (LLPS) in specific regions of the cell (Brangwynne et al, 2009; Zhao and Zhang, 2020). Intracellular physiological function disorder is a key event of tumorigenesis (Tong et al, 2022; Wang et al, 2020b; Zhang et al, 2022). Many tumor-related proteins, which exist intrinsic disorder regions, play key roles in promoting tumorigenesis and development (Iakoucheva et al, 2004;

[1]Department of Obstetrics and Gynecology, The First Affiliated Hospital of Chongqing Medical University, Chongqing, China. [2]Institute of Basic Medical Sciences, Hubei University of Medicine, Shiyan, China. [3]Department of Obstetrics and Gynecology, The Third Affiliated Hospital of Chongqing Medical University, Chongqing, China. [4]Department of Rheumatology and Immunology, Children's Hospital of Chongqing Medical University, National Clinical Research Center for Child Health and Disorders, Ministry of Education Key Laboratory of Child Development and Disorders, Chongqing Key Laboratory of Child Infection and Immunity, Chongqing, China. [5]College of Pharmacy, Chongqing Medical University, Chongqing, China. [6]Department of Pathology, Southwest Hospital, Third Military Medical University (Army Medical University), Chongqing, China. [7]Department of Gynecology and Obstetrics, Clinical Medical College and The First Affiliated Hospital of Chengdu Medical College, Chengdu, China. [8]Department of Pathology, School of Basic Medicine, Southern Medical University, Guangzhou, China. [9]These authors contributed equally: Hongyan Zhao, Qinglv Wei, Zhi Luo, Xiaoyi Liu. ✉E-mail: wangwei@cqmu.edu.cn; taoliu@hospital.cqmu.edu.cn; yiping@cqmu.edu.cn

Ukmar-Godec et al, 2019). For example, TAZ activates target gene transcription through phase separation (Lu et al, 2020). YTHDF protein promotes the formation of membrane-less granules in cells (Fu and Zhuang, 2020). YTHDF2 promotes the degradation of target genes by LLPS, and m⁶A-modified RNA promotes YTHDF2-mediated phase separation (Li et al, 2022; Ries et al, 2019; Wang et al, 2020a). Ovarian cancer has the highest mortality rate among gynecological malignancies, and the unclear pathogenesis of ovarian cancer is a key contributor to its high mortality rate (Cao et al, 2021; Feng et al, 2019; Siegel et al, 2022; Sung et al, 2021). However, biomolecular condensates as well as their link with m⁵C modification in ovarian cancer progression remain largely unknown.

In this study, we explored the role and potential mechanism of ALYREF in ovarian cancer. We found that ALYREF is abnormally upregulated in ovarian cancer and promotes ovarian cancer growth and migration through its m⁵C-modification recognition activity. ALYREF undergoes liquid-liquid phase separation in the nucleus and recognizes m⁵C-modified PARP10 mRNA. ALYREF interacts with PARP10 mRNA and promotes its stability by impeding the exosome complex. High expression of ALYREF is closely associated with worse prognosis for ovarian cancer patients. These results suggest that ALYREF may be a potential therapeutic target for ovarian cancer.

# Results

## ALYREF deficiency suppresses ovarian cancer growth and metastasis

To explore the role of ALYREF in ovarian cancer, we first analyzed ALYREF expression in ovarian cancer using the GEO database and CSIOVDB database (GSE54388, GSE66957, GSE18520, GSE40595, and GSE10971). The results showed that ALYREF was significantly upregulated in ovarian cancer compared with normal ovarian surface epithelial cells or fallopian tube epithelial tissues (Figs. 1A and EV1A). ALYREF was increased in high-grade serous ovarian cancer compared with normal ovarian surface epithelium in GSE27651 (Fig. EV1B). ALYREF was overexpressed in ovarian cancer in the CSIOVDB database (Fig. EV1C). Increased ALYREF protein expression was also observed in ovarian cancer in the CPTAC database and the Hu ovarian cancer cohort (Fig. 1B,C).

We next investigated the effects of overexpressing and knocking down ALYREF (shALY-1 and shALY-2) on the growth and migration of ovarian cancer cell lines, including A2780, OVCAR3, and SKOV3 cells (Fig. EV1D,E). ALYREF overexpression promoted the growth of ovarian cancer cells, as shown by CCK-8 and colony formation assays (Figs. 1D,F and EV1F). In contrast, silencing ALYREF inhibited the growth of ovarian cancer cell lines (Fig. 1E,G). Moreover, transwell assays demonstrated that ALYREF overexpression promoted the migration and invasion of ovarian cancer cells (Figs. 1H and EV1G), whereas ALYREF silencing impaired ovarian cancer cell migration and invasion abilities (Figs. 1I and EV1G).

To assess the effect of ALYREF in ovarian cancer in vivo, we used both subcutaneous xenograft and peritoneal metastatic models. To establish the subcutaneous xenograft model, ALYREF knockdown or control OVCAR3 cells were subcutaneously injected into the hind legs of mice. Three weeks later, the mice were euthanized and tumors were harvested for analysis (Fig. 1J). The tumor volumes and weights in the ALYREF knockdown group were significantly reduced compared with tumors in the control group (Fig. 1K,L). Ki-67 and Caspase-3 staining were used to examine the proliferation and apoptosis of tumor cells, respectively. Cell proliferation was reduced in the ALYREF knockdown group, while apoptosis was increased compared with the control group (Fig. EV1H,I). In the peritoneal metastasis model, the number of metastatic nodules was reduced in the knockdown group compared with the control group (Fig. 1M,N). Together, these results demonstrate that ALYREF promotes the proliferation and metastasis of ovarian cancer cells in vivo.

## ALYREF promotes the growth and metastasis of ovarian cancer cells in an m⁵C-dependent manner

ALYREF acts as an m⁵C-binding protein to regulate gene expression. Therefore, we examined whether ALYREF promoted tumorigenesis in ovarian cancer in an m⁵C-dependent manner. ALYREF specifically binds to m⁵C sites within mRNAs via K171 in its m⁵C recognition domain (Yang et al, 2017) (Figs. 2A and EV2A). Thus, we introduced a single point mutation, K171A, in a FLAG-tagged ALYREF (ALY-mut) plasmid. Ovarian cancer cells were transfected with the wild-type ALYREF (ALY-wt) or ALY-mut plasmids, and western blotting verified that both ALYREF proteins were effectively overexpressed in ovarian cancer cells (Fig. 2B). ALY-mut failed to promote proliferation in A2780 and OVCAR3 cells compared with ALY-wt (Fig. 2C–E). Furthermore, the migration and invasion capacities of A2780 and OVCAR3 cells expressing ALY-mut were decreased compared with cells expressing ALY-wt (Fig. 2F,G). These data indicate that the tumor-promoting effect of ALYREF in ovarian cancer relies on its binding to m⁵C-modified RNA.

To explore the interaction between ALYREF and m⁵C-modified RNA in ovarian cancer cells, we performed RIP-seq and RNA-BisSeq. We identified 1879 transcripts as ALYREF-binding transcripts (Fig. 2H; Dataset EV1), and functional pathway enrichment analysis of the transcripts revealed significant enrichment of signaling pathways such as ErbB, TNF, and cell cycle regulation (Fig. EV2B; Dataset EV2). Further analysis of the RNA-BisSeq data identified 7530 potential m⁵C modification sites in 1517 transcripts (Figs. 2I,J and EV2C; Dataset EV3). Among these m⁵C modification sites, 44.31% of the sites occurred in the coding sequence (CDS) regions, 25.11% occurred near start codons, and 14.03% occurred near 3′-UTRs (Fig. 2K,L). Most transcripts bound by ALYREF or with m⁵C modifications were protein-coding RNAs (Fig. EV2D). Functional enrichment analysis of ALYREF-binding transcripts with the m⁵C modification revealed an enrichment of tumor-associated pathways (Fig. 2M; Dataset EV4). Cumulation analysis showed that ALYREF was more prone to bind m⁵C-modified transcripts and the m⁵C modification in ALYREF-bound transcripts was more prone to occur in the CDS region compared with that in non-ALYRER bound transcripts (Figs. 2N and EV2E).

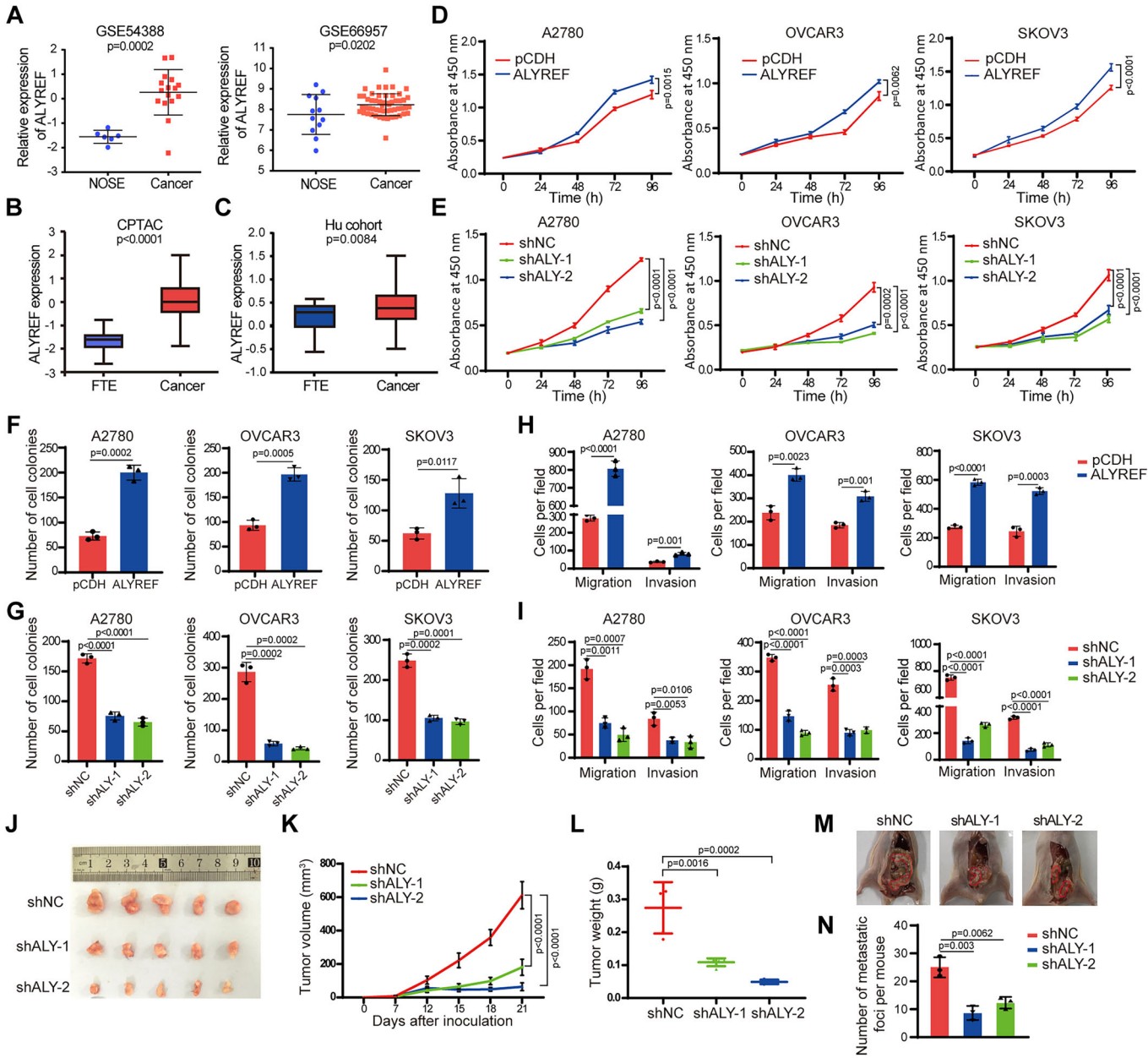

**Figure 1. Deficiency of ALYREF suppresses ovarian cancer cells growth and metastasis.**

(A) The expression of ALYREF in ovarian cancer and normal ovarian epithelium tissues in GEO datasets. GSE54388: NOSE ($n = 6$), Cancer ($n = 16$). GSE66957: NOSE ($n = 12$), Cancer ($n = 56$). (B, C) The protein level of ALYREF in ovarian cancer was analyzed according to the CPTAC database (FTE, $n = 25$, Cancer, $n = 100$) and the Hu cohort (FTE, $n = 23$, Cancer, $n = 83$). Box plots show the distribution of data across groups. The central line within the box represents the median value. The upper and lower edges of the box represent the 75th and 25th percentiles, respectively. The whiskers extend from the box to the maximum and minimum values within 1.5 times the interquartile range, with any data points beyond this range considered as outliers. (D) Cell growth was examined by CCK-8 assay in A2780, OVCAR3, and SKOV3 cells upon ALYREF overexpression. (E) Cell growth was examined by CCK-8 assays in ovarian cancer cells upon ALYREF knockdown. (F) Statistical results of colony formation assays in ovarian cancer cells upon ALYREF overexpression. (G) Numbers of colonies formed by ovarian cancer cells with or without ALYREF knockdown. (H) Quantification of migration and invasion transwell assays in A2780, OVCAR3, and SKOV3 cells with or without ALYREF overexpression. (I) Quantification of migration and invasion transwell assays in A2780, OVCAR3, and SKOV3 cells with ALYREF knockdown. (J–L) Control and ALYREF-depleted OVCAR3 cells were subcutaneously injected into BALB/C nude mice ($n = 5$). Tumors were isolated on day 21 after inoculation, and the tumor volumes and weights were measured. (M, N) Representative images and quantification of metastatic nodules formed by control and ALYREF-depleted ovarian cancer cells in the mouse peritoneal cavity ($n = 5$). $n = 3$ independent experiments (D–I). Data are shown as means ± S.D. P value was calculated by one-way ANOVA test with multiple comparisons (E, G, I, K, L, N) and unpaired two-sided Student's $t$ test (A–D, F, H). Source data are available online for this figure.

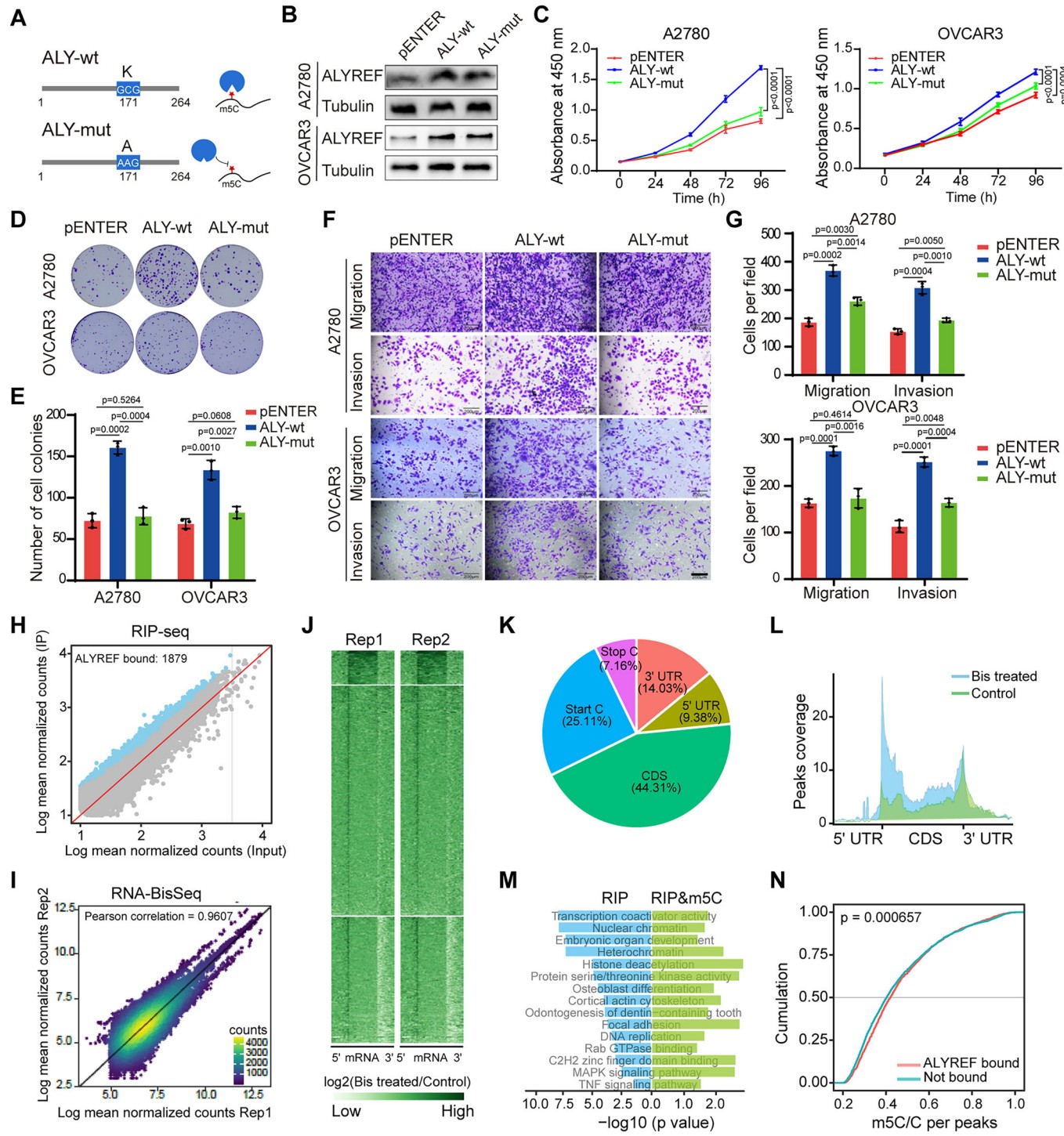

**Figure 2. ALYREF promotes the growth and migration of ovarian cancer cells dependently on m⁵C.**

(A) Schematic representation of wild-type (ALY-wt) and mutated (ALY-mut) ALYREF constructs. (B) Western blotting verifying ALYREF overexpression in A2780 and OVCAR3 cells. (C) CCK-8 assays detecting the proliferation of ovarian cancer cells transfected with wild-type ALYREF or K171A-mutant. (D) Colony formation of ovarian cancer cells transfected with wild-type ALYREF or K171A-mutant. (E) Numbers of colonies shown in (D). (F, G) Representative images and quantification of migration and invasion of ovarian cancer cells transfected with wild-type ALYREF or K171A-mutant. Scale bar, 200 μm. (H) Identification of ALYREF-binding transcripts by RIP-seq. (I) RNA-BisSeq identifying the transcripts with m⁵C modification. (J–L) Distribution of m⁵C modification on different regions of the transcripts. (M) Functional enrichment analysis of ALYREF-binding transcripts and m⁵C-modified transcripts. (N) The cumulative curve showing the association of ALYREF-binding transcripts with m⁵C modification. $n = 3$ independent experiments (C, E, G). Data are shown as means ± S.D. $P$ value was calculated by one-way ANOVA test with multiple comparisons (C, E, G) or a two-sided Mann–Whitney $U$ test (N). Source data are available online for this figure.

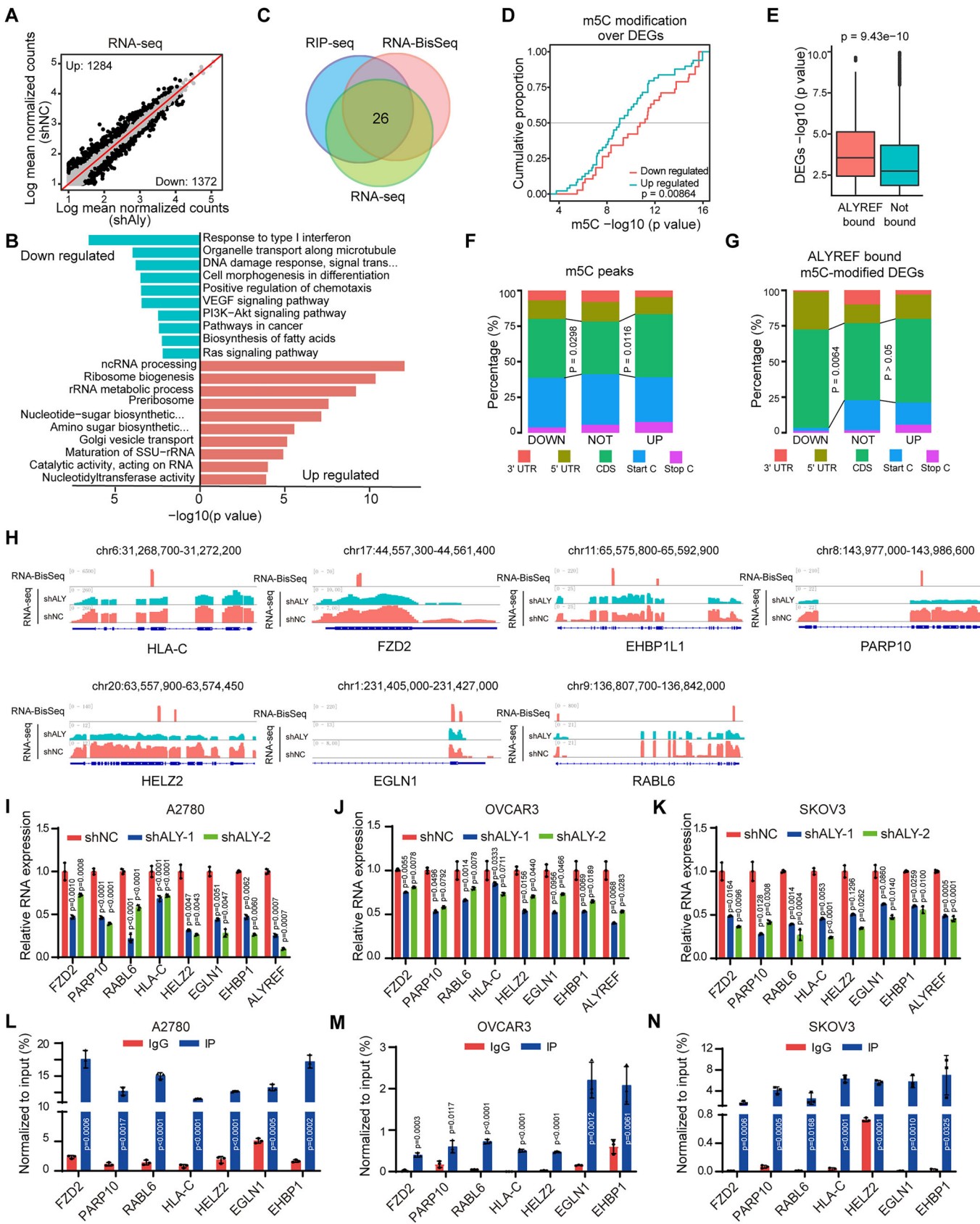

**Figure 3. Identification of the transcripts regulated by ALYREF through integrative multi-omics.**

(A) Differentially expressed genes (DEGs) induced by ALYREF knockdown in ovarian cancer cells. (B) Functional enrichment analysis of DEGs induced by ALYREF knockdown. (C) Overlapping analysis of data from RIP-seq, RNA-BisSeq, and RNA-seq. *P* value was calculated by "clusterProfiler" KEGG analysis. (D, E) Association analysis of DEGs with m5C modification and ALYREF binding, respectively. The central line within the box represents the median value. The upper and lower edges of the box represent the 75th and 25th percentiles, respectively. The whiskers extend from the box to the maximum and minimum values within 1.5 times the interquartile range, with any data points beyond this range considered as outliers. *P* value was calculated by a two-sided Mann–Whitney *U* test. (F, G) Distributions of m5C sites or m5C sites of ALYREF-binding transcripts on different regions of mRNA. *P* value was calculated by a Chi-square test. (H) Schematic diagram of IGV showing m5C peaks on the downregulated transcripts upon ALYREF knockdown. (I–K) RNA levels of ALYREF' targets were examined by RT-qPCR in ovarian cancer cells upon ALYREF knockdown. (L–N) RIP assays detecting the interaction between ALYREF and targeting mRNAs. *n* = 3 independent experiments (I–N). Data are shown as means ± S.D. *P* value was calculated by one-way ANOVA test with multiple comparisons (I–K) or unpaired two-sided Student's *t* test (L–N). Source data are available online for this figure.

## ALYREF regulates the expression of m5C-modified mRNAs

To further explore the genes regulated by ALYREF in ovarian cancer, RNA-seq was performed in A2780 cells with ALYREF knockdown. The results revealed that 1284 genes were upregulated and 1372 genes were downregulated upon ALYREF knockdown (Figs. 3A and EV3A; Dataset EV5). Gene Ontology (GO) analysis and the Gene Set Enrichment Analysis (GSEA) diagrams revealed that the downregulated genes were mainly enriched in cancer-related items (Figs. 3B and EV3B; Dataset EV6). The overlap between the RIP-seq, m5C-seq, and RNA-seq results revealed a total of 26 genes that may be potential targets of ALYREF (Fig. 3C). The cumulative curve analysis showed that m5C-modified transcripts tended to be decreased upon ALYREF knockdown and ALYREF-bound transcripts were also tended to be declined upon ALYREF knockdown, suggesting that ALYREF binds to m5C-modified transcripts and regulates their expression (Fig. 3D,E). Although significant differences in m5C distribution in the CDS region were observed in downregulated or upregulated transcripts compared with unchanged transcripts upon ALYREF knockdown, only downregulated transcripts bound by ALYREF tended to undergo m5C modification in their CDS regions (Fig. 3F,G). Among the 26 potential targets of ALYREF, 7 genes including PARP10 were downregulated upon ALYREF knockdown and mapped well to m5C sites (Fig. 3H). RT-qPCR confirmed downregulation of these genes in ovarian cancer cells upon ALYREF knockdown (Fig. 3I–K). RIP assays showed that target mRNAs were significantly enriched by ALYREF (Figs. 3L–N and EV3C).

## ALYREF outcompetes exosome complex component MTR4 in binding of m5C-modified PARP10 mRNA, enhancing its stability

Functional enrichment analysis of the differentially expressed genes upon ALYREF knockdown in RNA-seq showed that ALYREF was involved in the PI3K-AKT signaling pathway (Fig. 3B). PARP10 has been demonstrated to be involved in cancer progression via regulation of the PI3K-AKT signaling pathway (Zhou et al, 2022). Our RT-qPCR results also showed that PARP10 was significantly downregulated upon ALYREF depletion in ovarian cancer cells (Fig. 3I–N). The PARP/ARTD family includes 17 protein-modifying enzymes that share a conserved catalytic domain

containing ADP-ribosylation activity (Gibson and Kraus, 2012; Murthy et al, 2018). PARP10 promotes cell proliferation and tumorigenesis in cervical cancer cells by alleviating replication stress (Schleicher et al, 2018). Moreover, previous analysis of the TCGA database revealed that PARP10 expression is increased in 19% of breast tumors, and its overexpression facilitates cell proliferation and tumor progression (Khatib et al, 2022; Schleicher et al, 2018). Western blotting verified that PARP10 protein was significantly decreased in ovarian cancer cells following ALYREF depletion (Fig. 4A,B). Thus, we speculated that ALYREF regulated the PI3K-AKT signaling pathway through PARP10. To test this, we detected the activity of PI3K and AKT upon ALYREF depletion or PARP10 depletion, and the results revealed that either ALYREF depletion or PARP10 depletion significantly decreased the phosphorylation levels of PI3K and AKT in ovarian cancer cells (Fig. EV4A,B).

ALYREF is a nuclear junction factor that regulates mRNA export (Chi et al, 2013), and therefore we hypothesized that ALYREF might affect the nuclear export of PARP10 mRNA. To test this, we performed nuclear and cytoplasmic separation assays and examined the nuclear and cytoplasmic levels of PARP10 mRNA by qPCR. The results showed that PARP10 mRNA was blocked in the nucleus in ALYREF knockdown cells (Fig. 4C,D). m5C RNA modification promotes the pathogenesis of bladder cancer by stabilizing mRNAs, and ALYREF stabilizes PKM2 mRNA by binding m5C sites in the 3′UTR in bladder cancer (Wang et al, 2021; Wang et al, 2023). Thus, we next assessed the effect of ALYREF on the stability of PARP10 mRNA in cells treated with actinomycin D. The results indicated that knocking down ALYREF decreased the stability of PARP10 mRNA in ovarian cancer cells (Fig. 4E). The exosome complex is responsible for the processing and degradation of nuclear RNAs, and MTR4 is an important part of the exosome complex (Fan et al, 2017). RIP assays showed that MTR4 did not bind PARP10 mRNA in ALYREF-expressing ovarian cancer cells (Fig. EV4C). In addition, the results of nucleoplasmic isolation showed that MTR4 did not affect the localization of PARP10 mRNA (Fig. EV4D). However, the interaction between MTR4 and PARP10 mRNA was significantly increased upon ALYREF knockdown (Fig. 4F). ALYREF deficiency did not affect MTR4 expression in ovarian cancer cells (Fig. EV4E,F). These results support a model by which ALYREF and MTR4 competitively bind to PARP10 mRNA to regulate its expression. To confirm this possibility, we knocked down MTR4 in ALYREF-depleted ovarian cancer cells (Fig. EV4E,F). Notably,

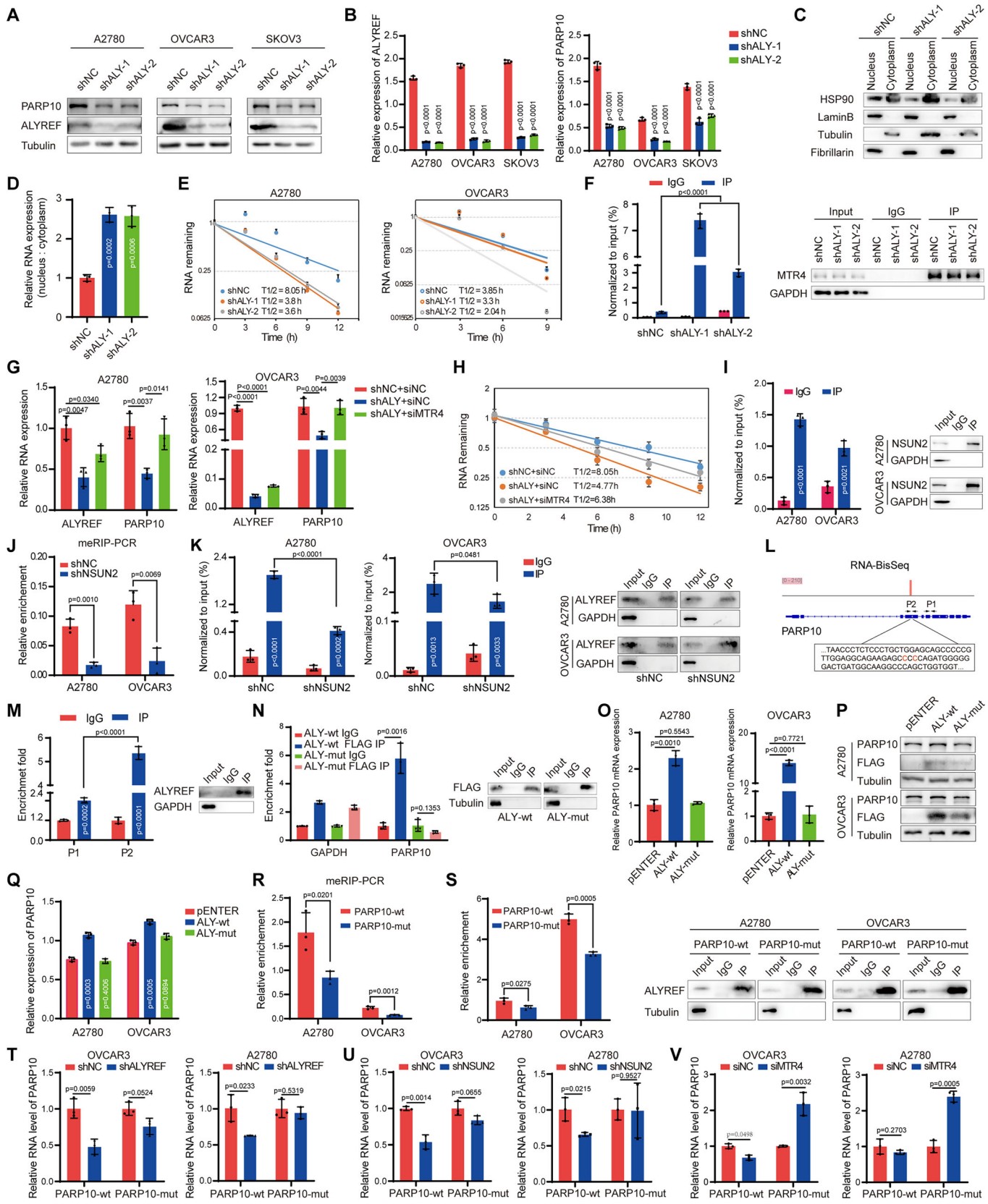

**Figure 4.   ALYRFE upregulates PARP10 expression in an m⁵C-dependent manner.**

(**A**) Western blotting assay detecting PARP10 expression in ovarian cancer cells upon ALYREF knockdown. (**B**) The protein expression was quantified by grayscale in (**A**). (**C**) Nucleocytoplasmic separation assays were confirmed by western blotting. (**D**) Effect of ALYREF deficiency on the nucleocytoplasmic ratio of PARP10 mRNA in ovarian cancer cells. (**E**) Effect of ALYREF deficiency on the stability of PARP10 mRNA in ovarian cancer cells. (**F**) Interaction between MTR4 and PARP10 mRNA in ovarian cancer cells with ALYREF knockdown or not by RIP and RT-qPCR assays. (**G**) Expression of PARP10 mRNA was assessed in ovarian cancer cells upon ALYREF knockdown alone or followed by MTR4 knockdown. (**H**) The stability of PARP10 mRNA was assessed in A2780 ovarian cancer cells upon ALYREF knockdown alone or followed by MTR4 knockdown. (**I**) RT-qPCR assays verifying the enrichment of PARP10 in NSUN2 RIP assays. (**J**) RT-qPCR assays detecting the enrichment of PARP10 in meRIP assays. (**K**) RT-qPCR assays detecting the enrichment of PARP10 in ALYREF RIP assays in ovarian cancer cells upon NSUN2 knockdown. (**L**) Schematic showed the m⁵C modification site of PARP10 according to the RNA-BisSeq. (**M**) Primers used to detect the enrichment of RARP10 mRNA in eCLIP assays shown in (**L**). (**N**) The eCLIP experiments detecting the interaction between ALYREF and PARP10 mRNA. (**O**) Effect of wild-type ALYREF or mutant overexpression on the expression of PARP10 mRNA in ovarian cancer cells. (**P**) Western blotting verifying overexpression of ALY-wt or ALY-mut in A2780 and OVCAR3 cell lines. (**Q**) The expression of the protein was quantified by grayscale in (**P**). (**R**) RT-qPCR assays detecting the enrichment of PARP10 in meRIP assays. (**S**) RT-qPCR assays detecting the enrichment of PARP10 in RIP assays. (**T**) RT-qPCR assays detecting expression changes of PARP10 in ovarian cancer cells with overexpression of PARP10-wt or PARP10-mut upon ALYREF knockdown. (**U**) RT-qPCR assays detecting expression changes of PARP10 in ovarian cancer cells with overexpression of PARP10-wt or PARP10-mut upon NSUN2 knockdown. (**V**) RT-qPCR assays detecting expression changes of PARP10 in ovarian cancer cells with overexpression of PARP10-wt or PARP10-mut upon MTR4 knockdown. $n = 3$ independent experiments. Data are shown as means ± S.D. $P$ value was calculated by one-way ANOVA test with multiple comparisons (**B, D, F, G, O, Q**) or unpaired two-sided Student's $t$ test (**I–K, M, N, R, S, T–V**). Source data are available online for this figure.

MTR4 knockdown restored PARP10 mRNA but not protein expression in ALYREF-depleted ovarian cancer cells (Fig. 4G). Furthermore, ALYREF knockdown decreased the stability of PARP10 mRNA, while concomitant knockdown of MTR4 promoted PARP10 mRNA stability (Figs. 4H and EV4G).

NOP2/Sun RNA methyltransferase family member 2 (NSUN2) is the main methyltransferase in mRNA m⁵C modification and implicated in ovarian cancer (Liu et al, 2024). To further validate the methylation of PARP10, RIP assays showed that NSUN2 bound PARP10 mRNA in ovarian cancer cells (Figs. 4I and EV4H). The meRIP assay showed that m⁵C modification of PARP10 mRNA was reduced when NSUN2 was knocked down in ovarian cancer cells (Fig. 4J). RIP assay results also indicated that the binding ability of ALYREF to PARP10 mRNA was reduced when NSUN2 was knocked down (Figs. 4K and EV4I). In addition, NSUN2 knockdown decreased PARP10 expression in ovarian cancer cells (Fig. EV4J,K). Next, we explored the implication of the m⁵C modification on PARP10 regulation by ALYREF. RNA-BisSeq results indicated the m⁵C modification site was located in the CDS of PARP10 mRNA (Fig. 4L). eCLIP-PCR assays confirmed that ALYREF bound to a region near the m⁵C site of PARP10 mRNA (Figs. 4M and EV4L). RIP assays showed that the ALYREF mutant could not bind PARP10 mRNA (Fig. 4N). Moreover, over-expressing ALY-wt but not ALY-mut promoted PARP10 expression at both the RNA and protein levels in ovarian cancer cells (Fig. 4O–Q). The meRIP assay illustrated that m⁵C modification of PARP10 mRNA was decreased upon mutation of the m⁵C site on PARP10 mRNA (Fig. 4R). Consistently, mutation of the m⁵C site on PARP10 mRNA significantly abrogated its interaction with ALYREF (Fig. 4S). Furthermore, knockdown of ALYREF or NSUN2 decreased the expression of wild-type PARP10 mRNA but not the m⁵C site-mutated PARP10 mRNA, whereas knock-down of MTR4 increased the expression of the m⁵C site-mutated PARP10 mRNA but not wild-type PARP10 mRNA (Fig. 4T–V). RNA half-life assays showed that knockdown of ALYREF or NSUN2 decreased the stability of wild-type PARP10 mRNA but not the m⁵C site-mutated PARP10 mRNA (Fig. EV4M). These results suggest that ALYREF promotes PARP10 expression by regulating the nuclear export and stability of PARP10 mRNA in an m⁵C-dependent manner.

## ALYREF undergoes LLPS

To further explore the functions of ALYREF in ovarian cancer, we performed immunofluorescence staining for ALYREF in ovarian cancer cells and found that ALYREF formed condensates (Fig. 5A). The overall phase separation characteristics of ALYREF protein were predicted by the phase separation protein prediction tool PhaSePred (Chen et al, 2022) (Fig. 5B). IUPred2A (Erdos and Dosztanyi, 2020), FuzDrop (Hatos et al, 2022), PONDR (Will-kommen et al, 2018), PScore (Vernon et al, 2018), and catGRANULE (Hatos et al, 2022) were also used to predict the phase separation property of ALYREF and its intrinsic disorder regions (IDRs) which contribute to condensation. ALYREF contains two IDRs, indicating a high potential of phase separation (Figs. 5C and EV5A–C). To confirm the LLPS of ALYREF, we expressed and purified a tGFP-ALYREF fusion protein from *E. coli*. Time-lapse microscopy showed that the contacted ALYREF droplets fused to form a larger spherical droplet, supporting the high fluidity property of the ALYREF droplets (Fig. 5D).

To probe the factors that affect LLPS in vitro, we tested the effects of temperature, protein concentration, and salt concentra-tion on the LLPS of ALYREF. ALYREF protein solution was depolymerized by heating from 4 °C to 37 °C, and the sample gradually became clear. After cooling to 4 °C, the ALYREF protein solution became turbid again. This cooling-induced LLPS indicated a higher critical solution temperature phase transition that occurred when cooling below the critical temperature (Fig. EV5D). At a salt concentration under 50 mM, ALYREF formed liquid-like droplets at a protein concentration as low as 1 µM. Moreover, disassembly of ALYREF droplets was observed with an increase in salt concentration, suggesting that ALYREF phase separation is salt concentration-dependent (Fig. 5E,F). In fluorescence recovery after photobleaching (FRAP) experiments, the fluorescence signal of a tGFP-ALYREF fusion protein was almost completely recovered at 85 s after bleaching (Fig. 5G,H; Movies EV1 and EV2). We next constructed stable ovarian cancer cells overexpressed with ALYREF labeled with tGFP and conducted FRAP in vivo. The results showed that the recovery times of the OVCAR3 cell line and the A2780 cell line were about 105 s and 66 s, respectively (Figs. 5I,J and EV5E,F). We also examined the effect of m⁵C-modified RNA on ALYREF

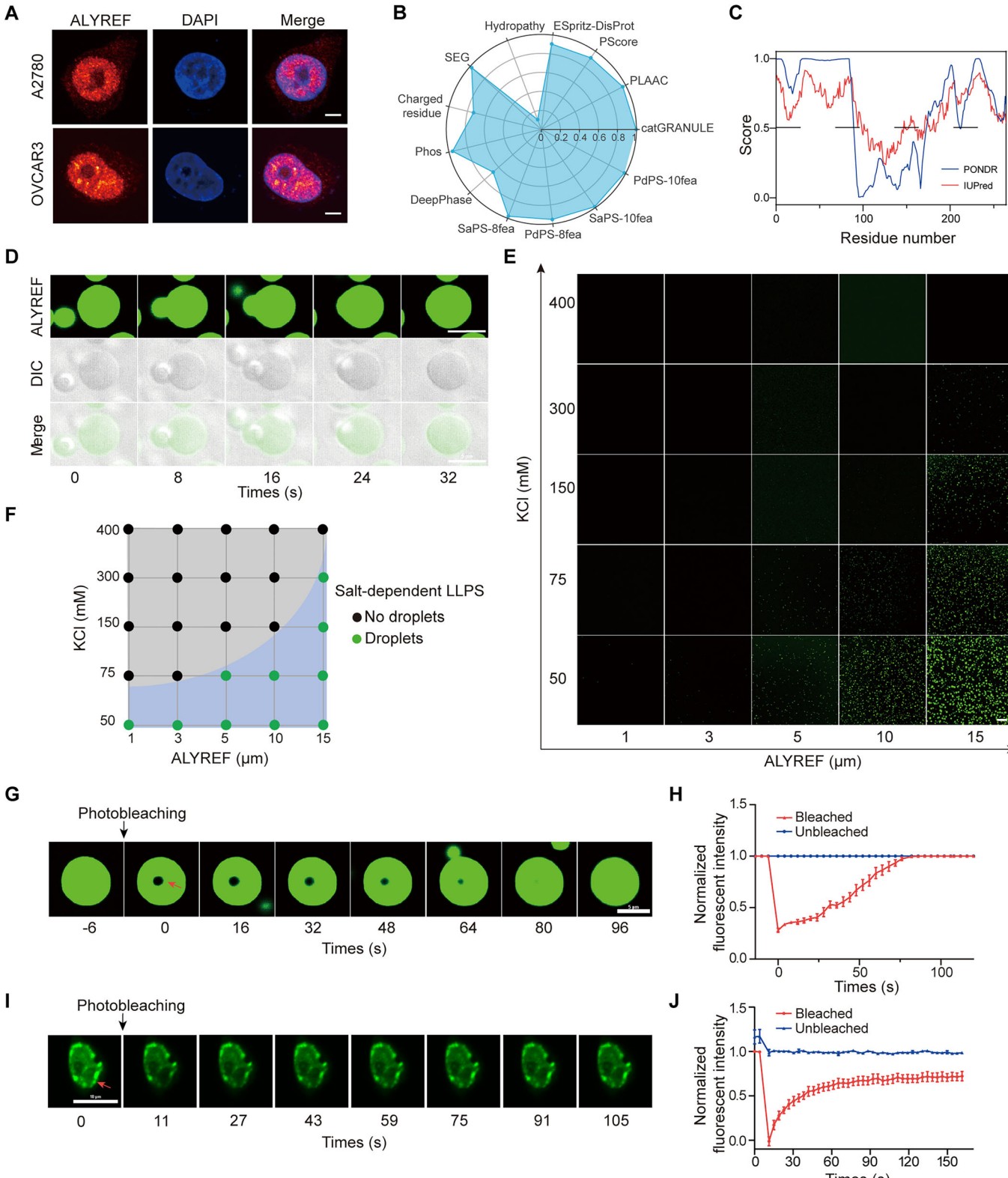

**Figure 5.   ALYREF undergoes liquid-liquid phase separation.**

(A) Immunofluorescence assays detecting the expression and localization of ALYREF in ovarian cancer cells. Scale bar, 5 μm. (B) The overall phase separation characteristics of ALYREF protein were predicted by PhaSePred. (C) The intrinsic disorder regions of ALYREF protein were predicted according to IUPred2A and PONDR. (D) In vitro protein fusion experiments of ALYREF tagged with GFP. Scale bar, 5 μm. (E) Effect of KCl on ALYREF phase separation. Scale bar, 5 μm. (F) Phase separation of ALYREF at different protein concentrations and salt concentrations. (G, H) The fluorescence bleaching recovery assays of ALYREF tagged with GFP were performed and the quantitative analysis. The black arrow represents the time point of photobleaching. The red arrow represents the photobleached region in the droplet ($n = 3$ independent experiments). Scale bar, 10 μm. (I, J) ALYREF forms droplet condensates in OVCAR3 cells and the quantitative analysis of condensates. The black arrow represents the time point of photobleaching. The red arrow represents the photobleached region in the droplet. In (I, J), the red curve represents the average of the normalized fluorescence intensities in different droplet photobleaching regions ($n = 3$ independent experiments). Source data are available online for this figure.

phase separation. ALYREF incubated with m⁵C-modified RNA resulted in larger areas of droplets compared with ALYREF incubated with non-m⁵C-modified RNA or without RNA, suggesting that the interaction with m⁵C-modified RNA may enhance the LLPS of ALYREF (Fig. EV5G). In addition, RNA fluorescence in situ hybridization assays showed that PARP10 mRNA was localized in ALYREF condensates in the nucleus of ovarian cancer cells (Fig. EV5H).

## ALYREF LLPS driven by IDRs is indispensable for ALYFER-mediated regulation of PARP10 expression and ovarian tumorigenesis

ALYREF contains three major domains: an N-terminal arginine-lysine-rich (R/K) domain, a central RNA-recognition motif domain (RRM), and a C-terminal arginine-lysine-rich (R/K) domain. ALYREF was predicted to contain IDRs in both the N-terminal and C-terminal domains (Fig. 6A). To map the key domains for its phase separation, we constructed the mutants of ALYREF. In vitro phase separation assays demonstrated that the N-terminal IDR and C-terminal IDR together drove LLPS of ALYREF, and ALYREF phase separation disappeared when the IDRs were absent. Moreover, mutation of Arg and Lys in IDRs to Ala also decreased ALYREF condensation (Fig. 6B). The aggregates were significantly smaller and PARP10 upregulation was disrupted in cells expressing the ALYREF mutants compared with cells with wild-type ALYREF expression (Fig. 6C,D). To study the regions where ALYREF performs phase separation, the opto-droplet system was constructed and photo induced IDR phase separation assays were carried out. The results show that the phase separation phenomenon of ALYREF occurs in the case of ALYREF full-length (Fig. 6E,F). To examine whether LLPS was involved in ALYREF regulation of PARP10 expression, we constructed GFP-ALYREF and its respective truncated forms in which IDRs were depleted or mutated and transfected the constructs into ovarian cancer cells. RT-qPCR showed that unlike wild-type ALYREF which promoted PARP10 expression, all the mutated forms of ALYREF exerted little effect on the PARP10 expression (Figs. 6G and EV6A). The nucleoplasmic isolation and RNA stability assays also showed that the mutated forms of ALYREF did not affect the nucleoplasmic distribution and stability of PARP10 mRNA (Figs. 6H,I and EV6B).

The mature ribonucleoprotein complexes (mRNPs) are recognized through multivalent interactions between the TREX subunit ALYREF and mRNP-bound exon junction complexes (EIF4A3 and MAGOH) (Asthana et al, 2022; Williams et al, 2018). To detect

whether ALYREF mutants with LLPS loss affected its interaction with other components in the TREX complex, we performed immunoprecipitation assays and found that only mutation of R/K to A destructed the interaction of ALYREF with EIF4A3 and MAGOH, whereas other LLPS-depleted ALYREF mutants were intactly bound to EIF4A3 and MAGOH. ALYREF interacts with other proteins primarily through amino acid sequences in the 55–182 region (Pacheco-Fiallos et al, 2023). These results suggest that LLPS depletion interferes with ALYREF-regulating PARP10 expression and is independent of its interaction with other components in the TREX (Fig. EV6C).

To confirm the effect of LLPS on ALYREF promoting ovarian cancer, truncations, mutants, and ALY-wt were overexpressed in ovarian cancer cells, respectively. ALY-wt overexpression promoted the proliferation, migration, and invasion abilities of ovarian cancer cells (Fig. 7A–C). However, LLPS depletion of ALYREF failed to facilitate ovarian cancer cell proliferation, migration, and metastasis (Fig. 7A–C). Consistently, in vivo assays also showed that LLPS depletion abrogated the promoting effect of ALYREF on tumor growth in mice (Fig. 7D–F). Together, these results demonstrate that ALYREF promotes PARP10 expression and tumorigenesis dependently on LLPS.

## Key functions of the ALYREF-PARP10 axis in ovarian cancer

To further examine the association of PARP10 with the tumor-promoting role of ALYREF in ovarian cancer, PARP10 was overexpressed in ALYREF-knockdown ovarian cancer cells (Fig. EV7A,B), and then cell growth and metastasis were assessed. ALYREF knockdown inhibited cell proliferation, whereas overexpressing PARP10 partly reversed this effect (Figs. 7G,H and EV7C). Furthermore, the inhibitory effect of ALYREF deficiency on cell migration and invasion was also reversed by forced expression of PARP10 (Figs. 7I and EV7D). These results indicate that PARP10 may be a key downstream target of ALYREF that facilitates ovarian cancer progression. The correlation between PARP10 and ALYREF RNA expression was analyzed in ovarian cancer tissues. The RNA abundance of ALYREF was positively correlated with PARP10 expression (Fig. 7J). To further investigate the effects of ALYREF and PARP10 on the development of ovarian cancer. Normal and ovarian cancer tissues were collected, and western blotting assays were performed. ALYRFE and PARP10 were highly expressed in ovarian cancer tissues (Fig. 7K). ALYREF expression was positively correlated with PARP10 expression in

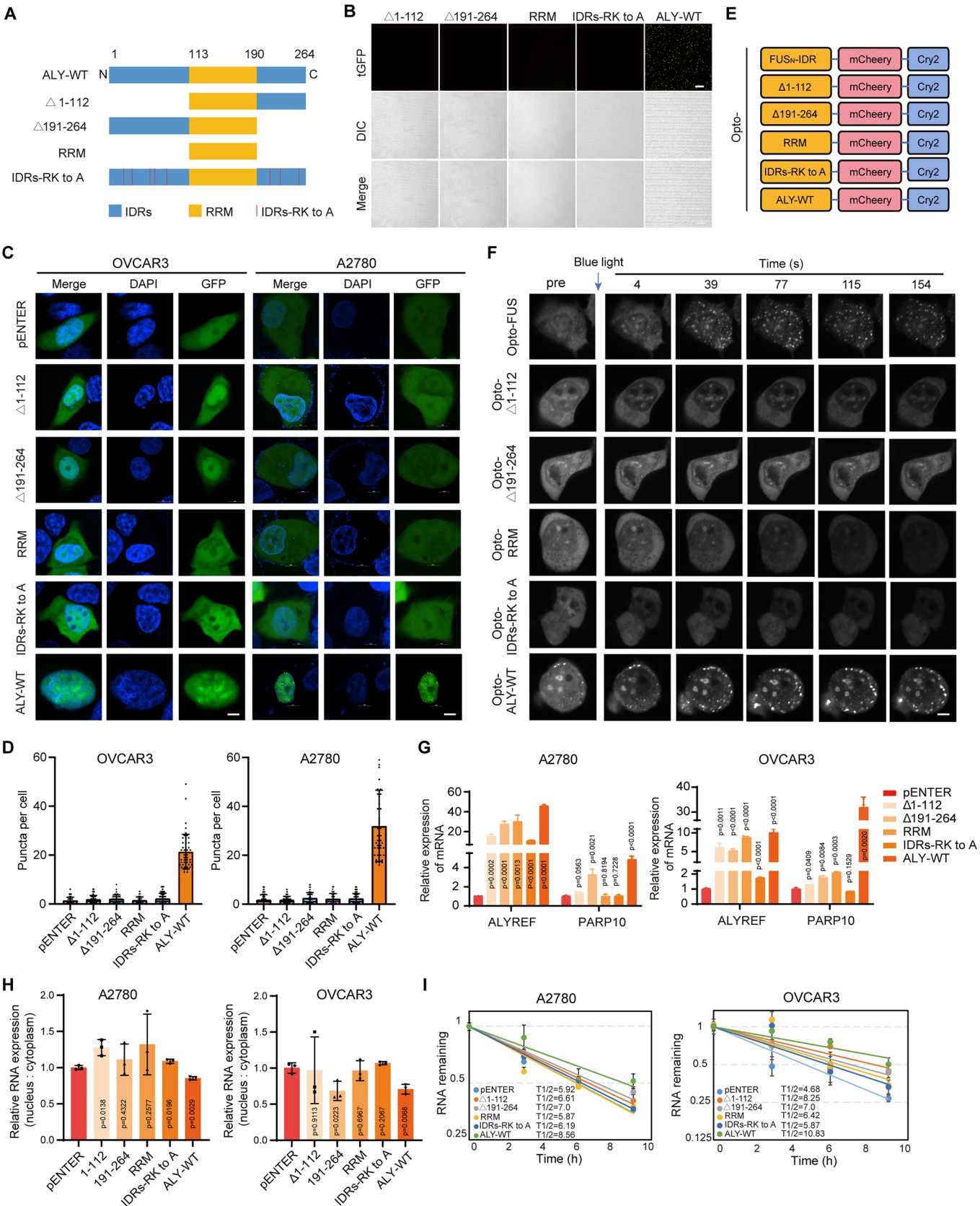

**Figure 6.   ALYREF LLPS was driven by arginine-lysine residues in IDRs.**

(A) Schematic diagram of the vector construction of ALYREF-wt, truncations, and mutants. (B) In vitro phase separation of ALYREF-wt, truncations, and mutants tagged with GFP was detected. Scale bar, 5 µm. (C) Representative images depicting exogenously expressed GFP-ALYREF and its respective truncated forms in OVCAR3 and A2780 cells are shown. A scale bar of 5 µm is provided for reference. (D) Statistical analysis of the number of LLPS puncta for each group ($n = 50$). (E) Schematic diagram of the opto-droplet system. (F) Representative images showing the clustering of ALYREF and its truncated form induced by blue light in HEK293T cells. Stimulation was achieved using a 488 nm laser. Scale bar: 5 µm. (G) RT-qPCR assays detecting PARP10 expression level in ovarian cancer cells with overexpression of ALYREF-wt, truncations, and mutants, respectively. (H) RT-qPCR assays detecting karyoplasmic distribution of PARP10 mRNA in ovarian cancer cells with overexpression of ALYREF-wt, truncations, and mutants, respectively. (I) Effect of ALYREF-wt, truncations, and mutants on the stability of PARP10 mRNA in ovarian cancer cells. $n = 3$ independent experiments (G–I). Data are shown as means ± S.D. *P* value was calculated by one-way ANOVA test with multiple comparisons (G, H). Source data are available online for this figure.

ovarian cancer (Fig. 7L). In addition, a similar positive association between ALYREF expression and PARP10 expression was observed in multiple cancers (Fig. EV7E).

## ALYREF is amplified and acts as a prognostic factor in ovarian cancer

Our results showed that ALYREF expression was increased in ovarian cancer. We then assessed the contribution of gene patterns including amplification and DNA methylation. We found that ALYREF was amplified in multiple cancers including ovarian cancer, and gene amplification was correlated with ALYREF expression, whereas DNA methylation was not significantly associated with ALYREF expression in ovarian cancer (Figs. 8A–C and EV8A,B). ALYREF expression in serous ovarian cancer at higher stages (III and IV) was significantly increased compared with expression in cancer at lower stages (I and II) (Figs. 8D and EV8C). Similar results were observed in ovarian cancer at higher grades (Fig. 8E). Moreover, high ALYREF expression was significantly associated with poor overall survival (OS), disease-free survival (DFS), and progression-free survival (PFS) in ovarian cancer patients (Fig. 8F,G). Consistently, higher PARP10 expression also predicted worse prognosis for ovarian cancer patients (Fig. EV8D). Higher expression of ALYREF portended a worse prognosis for ovarian cancer patients, regardless of p53 mutation status or tumor grades (Fig. EV8E,F). Furthermore, we analyzed the ALYREF expression in ovarian cancer through a tissue microarray including 134 ovarian cancer patients and found that ALYREF expression was significantly associated with poor survivals (Fig. EV8G,H). Notably, higher ALYREF expression was correlated with shorter survivals in patients at advanced stages (III and IV) (Fig. EV8H). Finally, we evaluated ALYREF expression as well as its prognostic values in pan-cancer. The results revealed that ALYREF was significantly increased and closely associated with poor OS in various cancers (Fig. EV8I–L). Together, these data indicate that increased expression of ALYREF predicts a poor prognosis for cancer patients.

## Discussion

ALYREF, a classic RNA-binding protein in the TREX multi-subunit RNA transport complex, plays a critical role in the export of mRNAs out of the nucleus (Chi et al, 2013). However, recent studies have demonstrated that ALYREF also regulates RNA stability in the nucleus including MYCN mRNA (Nagy et al, 2021). In this study, we found that ALYREF regulates global RNA abundance in ovarian cancer cells by binding transcripts in an m5C-dependent manner. Using a multi-omics strategy, we identified PARP10 as a critical target of ALYREF in ovarian cancer. ALYREF specifically recognizes m5C-modified PARP10 mRNA and enhances its stability. ALYREF is upregulated in ovarian cancer and is significantly associated with the poor survival of ovarian cancer patients.

Exosome complexes mediate the processing, decay, and degradation of cellular transcripts (Hartung et al, 2010; Mitchell et al, 1997; Sun et al, 2018). MTR4 is a conserved RNA helicase that performs functions together with the nuclear exosome and participates in the processing of structured RNAs, including the maturation of 5.8S rRNA and RNA degradation (Weir et al, 2010). RNA degradation ensures stable mRNA expression, helps eliminate abnormally redundant transcripts, and rebuilds the transcriptome when conditions change (Bresson et al, 2017; Pérez-Ortín et al, 2013; Sohrabi-Jahromi et al, 2019). In the nucleus, mRNA is primarily degraded from the 3′ to the 5′ end by the exosome complex and the MTR4 helicase (Kilchert et al, 2016; Wang et al, 2021). MTR4 binds with the cap-binding complex to regulate RNA degradation and nuclear export by competing with ALYREF (Fan et al, 2017). In this study, we examined the nuclear and cytoplasmic levels of PARP10 mRNA and found that PARP10 mRNA was increased in the nucleus following ALYREF knockdown. ALYREF knockdown decreased the stability of PARP10 mRNA, but concomitant MTR4 knockdown promoted PARP10 mRNA stability. Our data support a model where ALYREF and MTR4 competitively bind to PARP10 mRNA to regulate its expression.

The m5C recognition protein ALYREF specifically binds to m5C modification sites through K171 to promote the nuclear export of mRNAs (Yang et al, 2017). ALYREF also affects the occurrence and development of tumors by regulating the stability of mRNAs (Nagy et al, 2021; Wang et al, 2021). The m5C recognition protein YBX1 recognizes m5C-modified mRNA through the W65 indole ring in its cold shock domain and subsequently recruits ELAV like protein 1 (ELAVL1) to stabilize heparin-binding growth factor mRNA, which ultimately promotes bladder cancer cell proliferation and metastasis (Chen et al, 2019). YBX1 facilitates the maternal-to-zygotic transition by preventing maternal mRNA decay (Yang et al, 2019). The RNA methyltransferase TRDMT1 is recruited to DNA damage sites and promotes m5C modification, which in turn promotes homologous recombination (Chen et al, 2020).

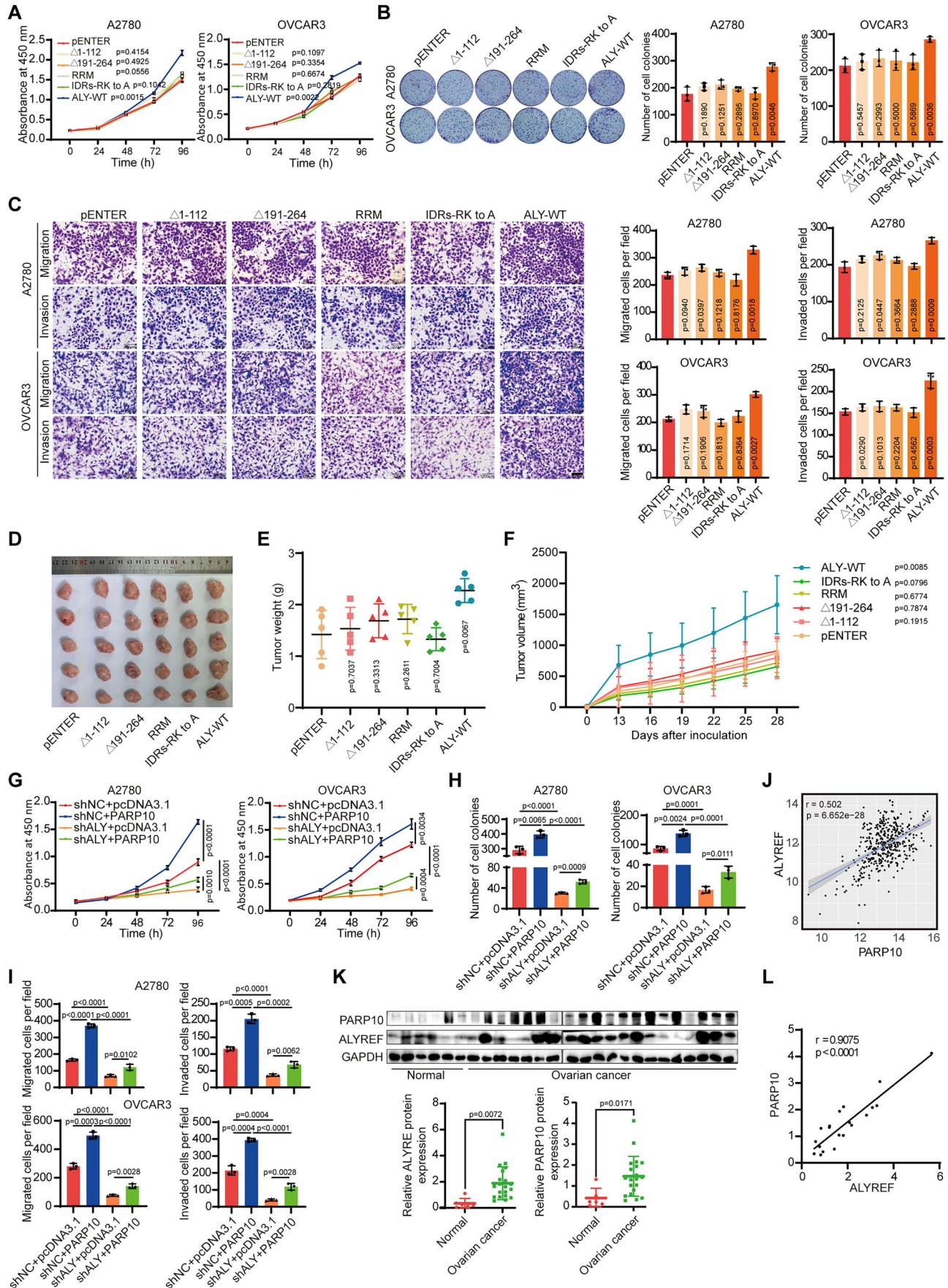

**Figure 7.  ALYREF promotes the growth and migration of ovarian cancer cells dependently on PARP10.**

(A) The proliferation of ovarian cancer cells with overexpression of ALYREF-wt, truncations, and mutants, respectively. (B) The colony formation assays of ovarian cancer cells with overexpression of ALYREF-wt, truncations, and mutants, respectively. (C) The migration and matrigel invasion assays of ovarian cancer cells with overexpression of ALYREF-wt, truncations, and mutants, respectively. Scale bar, 200 μm. (D) Representative images of xenograft tumors formed by ovarian cancer cells with overexpression of ALYREF-wt, truncations, and mutants, respectively in nude mice. (E) Assessment of the weight of xenograft tumors in nude mice ($n = 5$). (F) Assessment of the size of xenograft tumors in nude mice ($n = 5$). (G) The proliferation of ALYREF-depleted ovarian cancer cells with overexpression of PARP10. (H) The statistical result of colony formation of ALYREF-depleted ovarian cancer cells with overexpression of PARP10. (I) The statistical results of transwell migration and matrigel invasion of ALYREF-depleted ovarian cancer cells with overexpression of PARP10. (J) Spearman's rank correlation between ALYREF mRNA and PARP10 mRNA was carried out in ovarian cancer. (K) Expression levels of ALYREF and PARP10 in ovarian cancer tissues and normal fallopian tube epithelium by western blotting. (L) Correlation analysis of ALYREF and PARP10 in ovarian cancer tissues. $n = 3$ independent experiments (A–C, G–I). Data are shown as means ± S.D. P value was calculated by one-way ANOVA test with multiple comparisons (A–C, E, F, G–I) or unpaired two-sided Student's t test (K). Source data are available online for this figure.

A recent report using multimer analysis and RNA-BisSeq in HeLa cells showed that m5C methylation levels were inversely correlated with ribosome binding, suggesting that mRNA translation control may be negatively regulated by m5C modification (Schumann et al, 2020). m5C modification thus affects the maturation, stability, and translation of mRNA molecules and plays an important role in many physiological and pathological processes. We mutated the m5C binding site in ALYREF, and RIP results showed that ALY-mut could not bind to PARP10. ALYREF knockdown decreased the stability of PARP10 mRNA, but concomitant MTR4 knockdown promoted PARP10 mRNA stability. These data indicated that ALYREF specifically recognizes PARP10 mRNA through the K171 site and binds PARP10 mRNA by competing with MTR4. Through this mechanism, ALYREF affected the stability of PARP10 mRNA. Analyses of m5C modification at the whole transcriptome level have shown that m5C modification has species, tissue, and cell-type specificity. m5C modifications tend to be distributed in CDS, 3′ UTR, and CG-rich regions of mRNA (Chen et al, 2019; Yang et al, 2017). We found that the distribution of m5C sites in ovarian cancer cells was consistent with the known distribution patterns of m5C. The m5C modification site in PARP10 mRNA was located in its CDS.

RNA-binding proteins can regulate gene expression through phase separation, and RBP sequence characteristics contribute to its phase separation ability (Wiedner and Giudice, 2021). RBPs achieve weak non-covalent interactions through IDRs, RNA-binding domains, and dynamic post-translational modifications (PTMs) (Smith et al, 2020; Wang et al, 2018; Wiedner and Giudice, 2021). ALYREF is a classical RNA-binding protein. In vitro phase separation experiments and intracellular immunofluorescence observation of condensation showed that ALYREF-WT underwent phase separation. Phase separation is driven by weak polyvalent interactions between proteins usually including IDRs, low-complexity regions, RNA, and DNA (Boeynaems et al, 2019; Li et al, 2012; Ukmar-Godec et al, 2019). m5C-modified RNA promotes the phase separation of YBX2, which is involved in the regulation of RNA stability (Wang et al, 2022). Nascent mRNAs are processed and packaged into mRNP complexes and are recognized by the essential TREX for nuclear export (Singh et al, 2015). When TREX subunit ALYREF was truncated, there was no change in the interaction of ALYREF and TREX complex proteins (EIF4A3 and MAGOH). Only a mutation of R/K to A destroyed the interaction of ALYREF with EIF4A3 and MAGOH. ALYREF interacts with

other proteins primarily through amino acid sequences in the 55–182 region (Pacheco-Fiallos et al, 2023). These results suggest that TREX subunit ALYREF regulates PARP10 mRNA through phase separation. Notably, LLPS of ALYREF is critical for its regulation of m5C-modified PARP10 mRNA.

PARP10/ARTD10 belongs to the PARP/ARTD family, which exhibits ADP-ribosylation activity (Kleine et al, 2008). ADP-ribosylation is a unique post-translational modification that is involved in numerous cellular processes, including transcriptional regulation, signal transduction, cell death, metabolism, and DNA repair (Gibson and Kraus, 2012). Small molecule inhibitors of PARP are used to treat cancers with BRCA1 or BRCA2 mutations (McCann, 2019). PARP10/ARTD10 promotes cell proliferation, and PARP10 overexpression reduces the sensitivity of cells to replication stress and promotes the restart of stalled replication forks, which is important for relieving replication pressure (Nicolae et al, 2014). Xenotransplantation studies in mice have shown that PARP10/ARTD10 deficiency inhibits the growth of HeLa cells and that PARP10 overexpression promotes tumor progression (Schleicher et al, 2018). PARP10/ARTD10 mediates the single ADP ribosylation of GSK3β, which regulates cell proliferation, phosphorylates β-catenin, and induces the proteasomal degradation of β-catenin (Wu and Pan, 2010). PARP10 affects apoptosis, NF-κB signaling, and DNA damage repair (Dhoonmoon and Nicolae, 2023). Depletion of PARP10 impaired the PI3K-AKT and MAPK signaling pathways in oral squamous cell carcinoma (Zhou et al, 2022). In this study, we found that PARP10 was significantly overexpressed in ovarian cancer tissues and significantly related to the poor prognosis of patients. Knocking down PARP10 restrained the growth, migration, and invasion of ovarian cancer cells. Re-expressing PARP10 in ALYREF-depleted ovarian cancer cells rescued cell growth, proliferation, invasion, and migration to some extent. PARP10 or ALYREF knockdown decreased the phosphorylation level of PI3K and AKT in ovarian cancer cells. These data suggest that ALYREF plays a pro-tumor role in ovarian cancer, primarily by regulating PARP10 to participate in the PI3K-AKT signaling pathways. This work provides new ideas for studying the molecular mechanisms of ovarian cancer progression.

In this study, we report the function of ALYREF-mediated regulation of PARP10 mRNA stability by LLPS in ovarian cancer. Our findings enrich the mechanistic knowledge of how ALYREF regulates ovarian cancer progression via m5C, which may have significant application for new therapeutic strategies for ovarian cancer and other cancers (Fig. 8H).

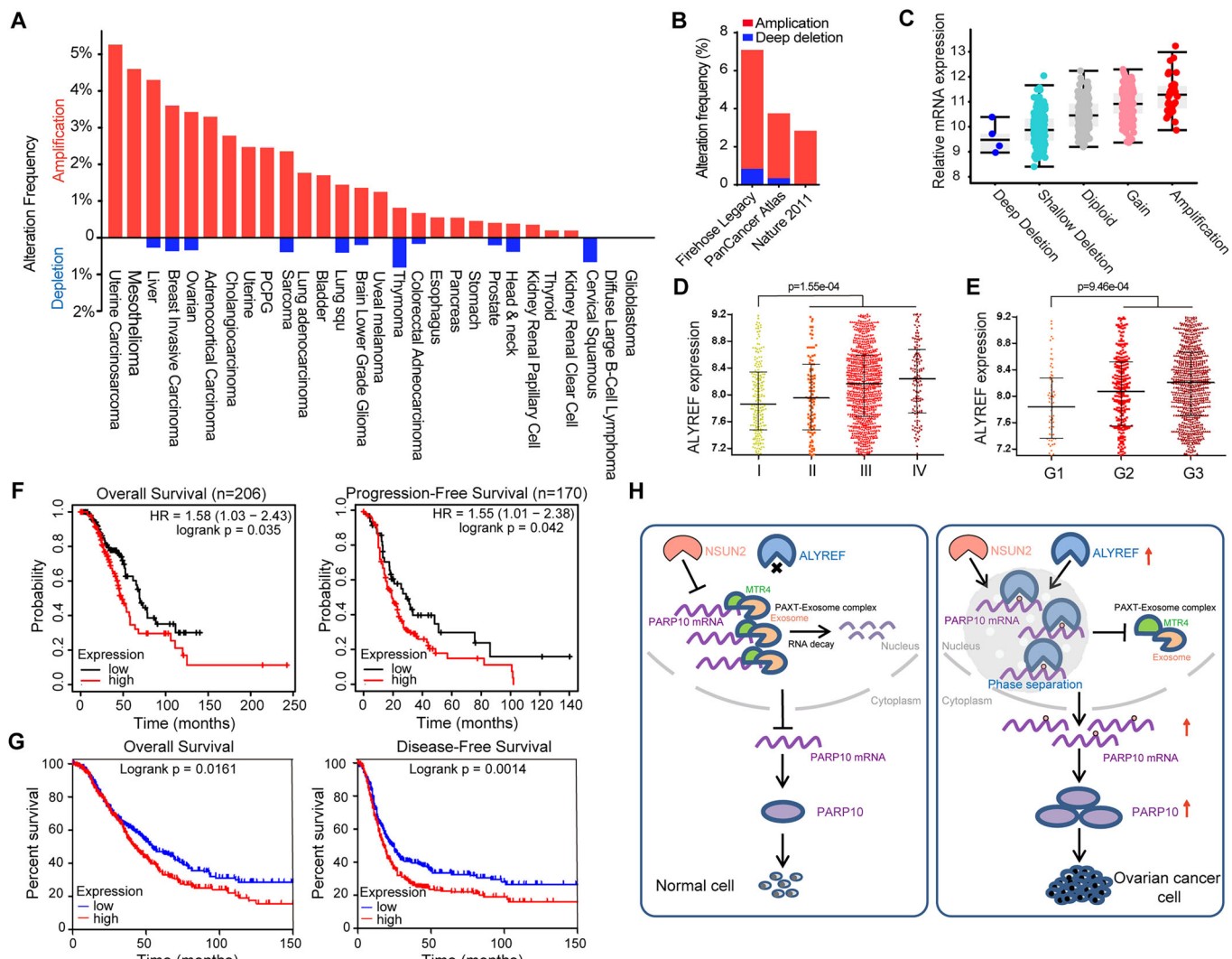

**Figure 8. ALYREF is amplified and predicts poor prognosis in cancers.**

(A) Alternation frequencies of ALYREF gene in various cancers including according to analysis in the TCGA pan-cancer database. (B) Alternation frequencies of ALYREF gene in ovarian cancers. (C) Analysis of the association between ALYREF amplification and its expression in ovarian cancer. The central line within the box represents the median value. The upper and lower edges of the box represent the 75th and 25th percentiles, respectively. The whiskers extend from the box to the maximum and minimum values within 1.5 times the interquartile range, with any data points beyond this range considered as outliers. (D, E) Expression of ALYREF in ovarian cancers at different FIGO stages and tumor pathological grades. (F, G) Associations of ALYREF expression with ovarian cancer patients' OS, DFS, and PFS. (H) Schematic illustrating the mechanisms of ALYREF-mediated stability of m5C-modified PARP10 mRNA by LLPS in ovarian cancer. Data are shown as means ± S.D. $P$ value was calculated by unpaired two-sided Student's $t$ test (D, E) two-sided Log-rank (Mantel-Cox) test (F, G). Source data are available online for this figure.

# Methods

### Reagents and tools table

| Reagent/resource | Reference or source | Identifier or catalog number |
|---|---|---|
| **Experimental models** | | |
| HEK293T | Chinese Academy of Medical Sciences | N/A |
| A2780 | Chinese Academy of Medical Sciences | N/A |
| OVCAR3 | Chinese Academy of Medical Sciences | N/A |

| Reagent/resource | Reference or source | Identifier or catalog number |
|---|---|---|
| SKOV3 | Chinese Academy of Medical Sciences | N/A |
| **Recombinant DNA** | | |
| pLKO.1 | Addgene | #10878 |
| psPAX2 | Addgene | #12260 |
| pMD2.G | Addgene | #12259 |
| shALY-1 | Self-construct | N/A |
| ShALY-2 | Self-construct | N/A |
| ShP10-1 | Self-construct | N/A |

| Reagent/resource | Reference or source | Identifier or catalog number |
|---|---|---|
| ShP10-2 | Self-construct | N/A |
| pcDNA3.1(+) | GENEWIZ Biotech | Custom order |
| PARP10-MYC | GENEWIZ Biotech | Custom order |
| pENTER | Vigene Biosciences | Custom order |
| pENTER-ALYREF-wt | Vigene Biosciences | Custom order |
| pENTER-ALYREF-mut | Vigene Biosciences | Custom order |
| pENTER-ALYREF△1-112 | Vigene Biosciences | Custom order |
| pENTER-ALYREF△191-264 | Vigene Biosciences | Custom order |
| pENTER-ALYREF_RRM | Vigene Biosciences | Custom order |
| pENTER-ALYREF_IDRs-RK to A | Vigene Biosciences | Custom order |
| pMAT-9S-tGFP(F) | Self-construct | N/A |
| pMAT-9S-tGFP-ALYREF | Self-construct | N/A |
| pMAT-9S-tGFP-ALYREF△1-112 | Self-construct | N/A |
| pMAT-9S-tGFP-ALYREF△191-264 | Self-construct | N/A |
| pMAT-9S-tGFP-ALYREF_RRM | Self-construct | N/A |
| pMAT-9S-tGFP-ALYREF_IDRs-RK to A | Self-construct | N/A |
| pCDH-CMV-MCS-EF1-copGFP | SBI | #CD511B-1 |
| ALYREF-pHR-mCh-Cry2WT | Addgene | #101221 |
| **Antibodies** | | |
| Tubulin | Proteintech | 11224-1-AP |
| GAPDH | Proteintech | 10494-1-AP |
| ALYREF | Proteintech | 16690-1-AP |
| PARP10 | Proteintech | 26072-1-AP |
| MTR4 | Proteintech | 12719-2-AP |
| FLAG-tag | Proteintech | 20543-1-AP |
| FLAG-tag | Proteintech | 66008-4-Ig |
| MYC-tag | Proteintech | 60003-2-Ig |
| HRP-conjugated Goat Anti-Rabbit IgG(H + L) | Proteintech | SA00001-2 |
| HRP-conjugated Goat Anti-Mouse IgG(H + L) | Proteintech | SA00001-1 |
| Goat anti-mouse IgG (H + L) DyLight 649 | Abbkine | A23610 |
| IgG | Millipore | 15006 |
| p-AKT | Absin | abs130002 |
| AKT | Absin | abs131789 |
| p-PI3K | Absin | abs130868 |
| PI3K | Absin | abs131198 |
| **Oligonucleotides and other sequence-based reagents** | | |
| pre-GAPDH-F | CATGGGTGTGAACCATGAGA | Custom order |
| pre-GAPDH-R | CAGGGGAGCGTGTCCATAG | Custom order |
| U6-F | CTCGCTTCGGCAGCACATATACT | Custom order |
| U6-R | ACGCTTCACGAATTTGCGTGTC | Custom order |

| Reagent/resource | Reference or source | Identifier or catalog number |
|---|---|---|
| GAPDH-F | TGCACCACCAACTGCTTAGC | Custom order |
| GAPDH-R | GGCATGGACTGTGGTCATGAG | Custom order |
| β-actin(F) | AGAGCTACGAGCTGCCTGAC | Custom order |
| β-actin(R) | AGCACTGTGTTGGCGTACAG | Custom order |
| 18 s rRNA(F) | CGATAACGAACGAGACTCTGGC | Custom order |
| 18 s rRNA(R) | CGGACATCTAAGGGCATCACA | Custom order |
| shALY-1(F) | CCGGGAACTCTTTGCTGAATTTGGACTCGAGTCCAAATTCAGCAAAGAGTTCTTTTTG | Custom order |
| shALY-1(R) | AATTCAAAAAGAACTCTTTGCTGAATTTGGACTCGAGTCCAAATTCAGCAAAGAGTTC | Custom order |
| shALY-2(F) | CCGGCGTGGAGACAGGTGGGAAACTCTCGAGAGTTTCCCACCTGTCTCCACGTTTTTG | Custom order |
| shALY-2(R) | AATTCAAAAACGTGGAGACAGGTGGGAAACTCTCGAGAGTTTCCCACCTGTCTCCACG | Custom order |
| shP10-1(F) | CCGGCGAGCTGCTCACTCTCTACTTCTCGAGAAGTAGAGAGTGAGCAGCTCGTTTTTG | Custom order |
| shP10-1(R) | AATTCAAAAACGAGCTGCTCACTCTCTACTTCTCGAGAAGTAGAGAGTGAGCAGCTCG | Custom order |
| shP10-2(F) | CCGGGCTCAGTTCCAGTGTGTCTTTCTCGAGAAAGACACACTGGAACTGAGCTTTTTG | Custom order |
| shP10-2(R) | AATTCAAAAAGCTCAGTTCCAGTGTGTCTTTCTCGAGAAAGACACACTGGAACTGAGC | Custom order |
| qFDZ2-(F) | GTGCCATCCTATCTCAGC | Custom order |
| qFDZ2-(R) | GAAACGCGTCTCCTCCTG | Custom order |
| qPARP10-(F) | AGGCGGCTGAGGAGTTTC | Custom order |
| qPARP10-(R) | GAGAAGGTGCTGCCCTTC | Custom order |
| m5Cq(PARP10-F) | ACCCTCTCCCTGCTGGAGCA | Custom order |
| m5Cq(PARP10-R) | GCCGAGTGCACCACCAGCTG | Custom order |
| qRABL6-(F) | GGGAGACAGGAACACG | Custom order |
| qRABL6-(R) | GCTCCAGTGGATGCTG | Custom order |
| qHLA-C-(F) | CATGAGGTATTTCGACAC | Custom order |
| qHLA-C-(R) | CTGTCGAACCGCACGAAC | Custom order |
| qHELZ2-(F) | GGTGCATCTGTGTCGTTTCC | Custom order |
| qHELZ2-(R) | CTGGCAGGATCTCAAAACTG | Custom order |
| qEGLN1-(F) | GAGAAGGCGAACCTGTAC | Custom order |
| qEGLN1-(R) | CAGCTTCAGCGCCGGCAG | Custom order |
| qEHBP1L1-(F) | TGCCACGGACGATGAC | Custom order |
| qEHBP1L1-(R) | GGCCATGAGCCTCATC | Custom order |
| qALY(F) | GCAGGCCAAAACAACTTCCC | Custom order |
| qALY(R) | AGTTCCTGAATATCGGCGTCT | Custom order |
| **Chemicals, enzymes and other reagents** | | |
| EcoRI | NEB | R0101S |
| AgeI | NEB | #R3552 |
| RPMI1640 | Gibco | 11879020 |
| DMEM | Gibco | 11965092 |
| penicillin/streptomycin | Gibco | 15140122 |
| FBS | VivaCell | C04001-500 |
| TRIzol | Invitrogen | 15596026CN |

| Reagent/resource | Reference or source | Identifier or catalog number |
|---|---|---|
| Immunoprecipitation cell lysis buffer | Beyotime | P0013 |
| protease inhibitor | Beyotime | P1006 |
| CCK-8 | Biosharp | BS350B |
| crystal violet | Sigma-Aldrich | C0775 |
| Matrigel | Corning | 354277 |
| Protein G beads | Invitrogen | 88848 |
| Actinomycin D | Sigma-Aldrich | A4262 |
| jetPRIME | Ployplus | 101000046 |
| PEI | Plosciences | #24765 |
| 4% paraformaldehyde | Biosharp | BL539A |
| **Software** | | |
| ImageJ | National Institutes of Health | https://imagej.net/downloads |
| GraphPad Prism | GraphPad Company | https://www.GraphPad.com |
| Adobe Illustrator | Adobe Systems Incorporated | https://www.adobe.com/products/illustrator.html |
| Adobe photoshop | Adobe Systems Incorporated | https://www.adobe.com/products/photoshop.htm |
| **Other** | | |
| BCA Protein Assay Kit | Solarbio | PC0020 |
| IF-FISH Kit | BersinBio | Bes1024M |
| Transwell | Corning | CLS3464 |

## Cell culture

The A2780, OVCAR3, and SKOV3 ovarian cancer cell lines and HEK293T human embryonic kidney cells were purchased from the Cell Center of the Chinese Academy of Medical Sciences (Beijing, China). All cell lines were stored in liquid nitrogen. A2780 and OVCAR3 were cultured in RPMI1640 medium with 10% FBS (VivaCell, China) and 1% penicillin/streptomycin (Gibco, Grand Island, NY, USA). SKOV3 and HEK293T cells were maintained in DMEM with 10% FBS (VivaCell, China). Cells were cultured at 37 °C in an incubator with 5% CO$_2$. Mycoplasma was removed using a mycoplasma removal kit (TranssafeTM Mycoplasma Elimination Reagent (TransGen Biotech, China)) to prevent cell contamination. The Mycoplasma detection kit (TransDetect® PCR Mycoplasma Detection Kit (TransGen Biotech)) was used for mycoplasma detection. Before conducting experiments, it was ensured that all cell lines were free of mycoplasma contamination.

## Plasmids

shRNA sequences were cloned into the pLKO.1 vector (#10878, Addgene) for lentivirus particle production. ALYREF knockdown plasmids were termed shALY-1 and shALY-2, PARP10 knockdown plasmids were termed shP10-1 and shP10-2, and the control lentivirus was termed NC. The ALYREF coding sequence was cloned into the pCDH-CMV-MCS-EF1-copGFP vector (#CD511B-

1, SBI). The pENTER-ALYREF-wt (ALYREF-FLAG) and pENTER-ALYREF-mut (K171A) plasmids were purchased from Vigene Biosciences Co. PARP10-MYC and mutation sequences were cloned into the pcDNA3.1(+) vector (GENEWIZ Biotech, China).

For lentivirus production and infection, pCDH or shRNA and packaging vectors psPAX2 (#12260, Addgene) and pMD2.G (#12259, Addgene) were transfected into HEK293T cells via transfection reagent (Polyplus, France). After 48 h, the culture medium was filtered with 0.22 μm PVDF filters to collect the lentivirus particles. Lentivirus infection was performed in cultured cells at 30% confluence.

For plasmid transfection, ALYREF-wt, ALYREF-mut, PARP10-wt, and PARP10-mut expression plasmids were transiently transfected into ovarian cancer cells using the transfection reagent (Polyplus) when the cell confluence reached 80%.

## RNA isolation and qPCR

Total RNA was extracted from A2780, OVCAR3, and SKOV3 ovarian cancer cells using TRIzol reagent (Invitrogen, Waltham, MA, USA). The qPCR primers were designed by the primer bank website (https://pga.mgh.harvard.edu/primerbank/) and synthesized at TSINGKE (China). RNA was reverse transcribed to cDNA using the Reverse Transcription Kit (Vazyme, China). qPCR was performed using SYBR Green Master Mix (Vazyme, China) on a Bio-Rad CFX96 real-time PCR system (Bio-Rad, Hercules, CA, USA) to detect mRNA expression levels. GAPDH mRNA or 18 s rRNA was used as the internal control for qPCR, and the relative expression value of each gene was calculated using the $2^{-\triangle\triangle CT}$ method.

## Western blotting

Cells were washed with PBS, resuspended in western and immunoprecipitation cell lysis buffer (Beyotime, China) with protease inhibitor (Beyotime, China), and incubated on ice for 30 min. The samples were subjected to freezing centrifugation at 12,000× g/min for 15 min and the supernatants were collected. Protein concentration was measured using the BCA Protein Assay Kit (Solarbio, China). Protein samples were separated by SDS-PAGE and transferred onto 0.22 μm PVDF membranes (Millipore, Burlington, MA, USA). The membranes were blocked in 5% non-fat milk in TBST and incubated with primary antibodies: anti-Tubulin, anti-GAPDH, anti-ALYREF, anti-PARP10, anti-MTR4, anti-FLAG-tag, and anti-MYC-tag (all from Proteintech, China). After washing, membranes were incubated with anti-mouse and anti-rabbit secondary antibodies (Proteintech). Protein bands were detected using Immobilon Western Chemiluminescent HRP Substrate (Millipore).

## Cell growth and proliferation assays

Cells were collected and counted with a Countstar Automated Cell Counter (China). For growth assays, 2000 or 2500 cells were inoculated into 96-well plates and cultured at 37 °C in a 5% CO$_2$ incubator for 0, 24, 48, 72, and 96 h. After adding 10% CCK-8 solution (Biosharp, China), the plates were incubated for another 2 h in the incubator. Cell viability was determined by measuring absorbance at 450 nm. For proliferation assays, 2000 or 3000 cells

were inoculated into 6-well plates and cultured for 10 days at 37 °C in a 5% $CO_2$ incubator. The cells were fixed with 4% paraformaldehyde for 30 min, stained with crystal violet (Sigma-Aldrich, St. Louis, MO, USA) for 1 h, and washed twice with PBS. The number of cell clones was counted using ImageJ software.

## Transwell migration and invasion assays

A 24-well Transwell chamber system (Millipore) was used to evaluate the migration and invasion ability of ovarian cancer cells. Cells were collected and counted with a Countstar Automated Cell Counter. Cells ($1 \times 10^5$) were resuspended in 0.2 mL of serum-free medium and seeded into the upper chamber; the lower chamber was filled with 0.6 mL of culture medium with 20% serum. For invasion assays, the upper chamber was coated with Matrigel (Corning, NY, USA) before plating cells. The chambers were then incubated at 37 °C for 24 h. Cells were then fixed with 4% paraformaldehyde for 30 min, stained with crystal violet (Sigma-Aldrich) for 1 h, and washed with PBS. Non-migrating or non-invasive cells in the upper chamber were wiped off with a cotton swab. Migrated or invaded cells were imaged under a ×20 microscope and counted using ImageJ software.

## Tumor tissues

The ovarian cancer and ovarian epithelial tissue samples were collected from female patients aged from 31 to 81 and underwent surgery at Army Specialty Medical Center and the Third Affiliated Hospital of Chongqing Medical University. All patients provided written informed consent and all tissue samples were confirmed by pathological examination. The relevant experiments were approved by the Third Affiliated Clinical Hospital of Chongqing Medical University (202125) and the Ethical Review Committee of Army Specialty Medical Center (2022374) and conducted in accordance with the Declaration of Helsinki.

## Immunohistochemistry (IHC)

The formalin-fixed paraffin-embedded tissues were dewaxed and rehydrated, and then incubated with specific primary antibodies and the corresponding secondary antibody after antigen retrieval and blocking. With diaminobenzidine staining, the final score of the microarrays was evaluated independently by two pathologists. The composite score was determined by using the H-score system, in which staining intensity was graded as 0, 1, 2, and 3, and the percentage of positive cells was categorized as 0%: 0, 1% to 25%: 1, 26% to 50%: 2, 51% to 75%: 3, and >75%: 4. The final composite score was obtained by multiplying the intensity and extent scores.

## In vivo tumorigenesis and metastasis assays

For tumorigenesis assays, ALYREF knockdown or control OVCAR3 cells ($1 \times 10^7$) in PBS were injected into the hindlimb of 4-week-old BALB/c nude mice. The weights and volumes of tumors were measured each week or every 3 days. After 3 weeks, the mice were euthanized, and the tumor weights were recorded. Collected tumor tissues were fixed with 4% paraformaldehyde for IHC analysis (Servicebio, China). For metastasis assays, $5 \times 10^6$ cells were injected into the abdominal cavity of BALB/c nude mice. After

3 weeks, the mice were euthanized, and the number of metastatic nodules was counted. Specific pathogen-free (SPF) experimental animals were housed in a barrier-sustained housing facility compliant with the SPF grade standards according to protocols approved by the Institutional Animal Care and Use Committee of Hubei University of Medicine (No. 00300229).

## RNA immunoprecipitation and high-throughput sequencing (RIP-seq)

Approximately $1 \times 10^7$ cells were harvested, washed with PBS, and resuspended in 0.5 mL IP lysis buffer (20 mM HEPES, 150 mM NaCl, 0.5 mM EDTA, 10 mM KCl, 0.5% NP-40, 10% glycerol, 1.5 mM DTT, 1 mM PMSF, RNase inhibitor (10 U/mL)). The cells were lysed on ice for 30 min, and the lysates were centrifuged at 12,000 rpm/min for 15 min. The supernatants were collected and incubated with anti-ALYREF (5 μg) or control IgG (5 μg) antibody and 50 μL of Protein G beads (Invitrogen, Carlsbad, CA, USA) at 4 °C for 12 h. The mixtures were washed three times with wash buffer, and the RNA was extracted for library construction and sequencing.

## RNA bisulfite genomic sequencing (RNA-BisSeq)

Approximately $1 \times 10^7$ cells were collected, washed with PBS, and total RNA was isolated using TRIzol reagent (Invitrogen). Poly(A)-modified RNAs were subsequently purified using the PolyTract mRNA Isolation System (Promega, Madison, WI, USA). The Bis-($m^5C$)-mRNA was prepared using the MethylCode Bisulfite Conversion Kit (Invitrogen) following the manufacturer's instructions. Both the BisSeq-mRNA and Mock-mRNA samples were sent to the Cloudseq company (China) for library construction and sequencing in duplicate.

## RNA sequencing (RNA-seq)

Total RNA was isolated from control and ALYREF knockdown A2780 cells using TRIzol reagent (Invitrogen). Poly(A)-modified RNAs were subsequently purified using the PolyTract mRNA Isolation System (Promega) for cDNA library construction. All samples were sequenced by the Illumina HiSeq X platform (ANOROAD, China). Each group was sequenced in duplicate.

## Enhanced UV crosslinking immunoprecipitation

Approximately $1 \times 10^7$ ALYREF-overexpressing cells were prepared for immunoprecipitation experiments and eCLIP-qPCR. UV crosslinking was conducted at the wavelength of 254 nm at 150 mJ. The crosslinked cells were collected, lysed in lysis buffer, digested with RNase I, and fragmented with DNase. The cell lysate was incubated with a specific antibody or control IgG and 50 μL of Protein G beads (Invitrogen) at 4 °C for 12 h. The RNA-binding protein compound was analyzed by western blotting. The RNA was isolated by TRIzol reagent (Invitrogen) for eCLIP-qPCR.

## RNA stability assay

Cells were plated into a 12-well plate and cultured in 5 μg/mL actinomycin D (Sigma-Aldrich) at 37 °C for 0, 3, 6, 9, and 12 h.

Total RNA was extracted using TRIzol reagent (Invitrogen). RNA expression was analyzed by qPCR.

## Immunofluorescence (IF)

The cells were fixed with 4% paraformaldehyde, permeabilized with 0.2% Triton X-100 in PBS, blocked with 1% BSA in PBST, and incubated with an anti-ALYREF antibody (1:200, Abcam) overnight at 4 °C. Then, the cells were washed with PBS, incubated with the secondary antibody (1:200, goat anti-mouse IgG (H + L) DyLight 649) for 1 h, incubated with DAPI for 10 min, and washed with PBS for 30 min. The cells were imaged under a laser-scanning confocal microscope (OLYMPUS).

## RNA fluorescence in situ hybridization (RNA-FISH)

The cells were fixed with 4% paraformaldehyde and treated following the protocol of the IF-FISH Kit (BersinBio). The antibodies are as follows: anti-ALYREF (1:200, Abcam) and secondary antibody (1:200, goat anti-mouse IgG (H + L) DyLight 649). The images were acquired with a laser-scanning confocal microscope (OLYMPUS).

## Opto-droplets assay

ALYREF and its truncated form were cloned into pHR-mCh-Cry2WT (Addgene #101221). HEK293T cells were plated at 70% confluence 24 h prior to transfection. Plasmids were transfected into the cells using PEI (Plosciences #24765), and after 48 h, the transfected cells were transferred to confocal dishes for subsequent experiments. Imaging was performed using the Nikon A1R confocal microscopy, employing a ×63 oil objective (zoom = 4). Cells were visualized using two laser wavelengths (488 nm for Cry2 activation and 561 nm for mCherry imaging). Imaging with the 561 nm laser was conducted prior to fluorescent stimulation. Droplet formation was induced with 488 nm light for 10 s, followed by capturing images every 4 s under 561 nm light.

## Vector construction, protein expression, and purification assays

The plasmids (ALYREF, ALYREF△1-112, ALYREF△191-264, ALYREF_RRM, and ALYREF_IDRs-RK to A) were separately cloned to pENTER and pMAT-9S-tGFP vectors. The ALYREF-WT, truncation, and mutant proteins were expressed in and purified from *E. coli* BL21 cells (Liu et al, 2021). The plasmids containing 6×His-, MBP-, and tGFP-genes were transformed into *E. coli* BL21 competent cells. Cells were heat incubated, spread plated, and incubated overnight at 37 °C. Single colonies were amplified in 3 ml of LB medium containing 100 mg/mL ampicillin at 37 °C overnight on a shaker at 220 rpm/min. The bacterial culture suspension was then added to 1 L of LB medium containing 100 mg/mL ampicillin and sodium, and the culture was incubated at 37 °C on a shaker at 220 rpm/min. When the $OD_{600}$ was 0.6–0.8, 0.5 mM isopropyl-β-D-galactosidase (IPTG) was added, and the culture was incubated at 16 °C for 1 h. The medium was then cultured at 16 °C for 18 h. After centrifugation at 4 °C at 4000 rpm/min for 15 min, cell pellets were collected.

The method of 6×His-MBP-tGFP-ALYREF-WT protein purification was as follows: The cell precipitate was resuspended in a 40 ml MBP column binding buffer solution (50 mM Tris-HCl 8.5, 10 mM MgCl$_2$·6H$_2$O, 2 M KCl, 5% glycerol (v/v)) containing 1 mM DTT and 1 mM PMSF. The cells were lysed using a JN-02C Low-temperature Ultra-high Pressure Continuous Cell Crusher (JNBIO) at 4 °C at 1500 bar for five cycles and an ultrasonic cell crusher (SCIENTZ) at 4 °C for ten cycles (120 W, 10 min, on 15 s, off 30 s). The crude lysate was collected by centrifugation for 60 min at 20,000 rpm/min at 4 °C. The supernatants were separated by MBP column affinity chromatography (MBPTrapTM HP 5 ml), HP column affinity chromatography (HiTrapTM Heparin HP 5 ml), and gel filtration chromatography (SuperdexTM 200 Increase 10/300 GL) in order. The crude protein samples were transferred to a MBP column, bound with binding buffer (50 mM Tris-HCl 8.5, 10 mM MgCl$_2$·6H$_2$O, 2 M KCl, 5% glycerol (v/v)), and eluted with MBP column elution buffer (50 mM Tris-HCl 8.5, 10 mM MgCl$_2$·6H$_2$O, 50 mM maltose, 2 M KCl, 5% glycerol (v/v)); elution peak samples were collected. The protein samples were concentrated to 500 μl at 4 °C at 3000× $g$/min using Amicon Ultra-15 Centrifugal Filters Ultracel-50k (Merck Millipore). The protein samples were diluted to 5 mL with HP column storage buffer (20 mM HEPES-HCl 7.5, 1 mM DTT, and 10% glycerol (v/v)). The protein samples were transferred to a HP affinity chromatography column, bound with binding buffer (20 mM HEPES-HCl 7.5, 1 mM DTT, 100 mM KCl, and 10% glycerol (v/v)), and eluted with HP affinity chromatography column elution buffer (20 mM HEPES-HCl 7.5, 1 mM DTT, 2 M KCl, and 10% glycerol (v/v)). The peak protein samples were concentrated to 500 μl at 4 °C at 3000× $g$/min using Amicon Ultra-15 Centrifugal Filters Ultracel-50k (Merck Millipore). The protein samples were eluted with gel filtration chromatography column elution buffer (50 mM Tris-HCl 7.4, 500 mM KCl, 1 mM DTT, and 5% glycerol (v/v)). The protein samples were further purified on a polyacrylamide gel, collected, concentrated, placed in 20 μl aliquots in PCR tubes, quick-frozen in liquid nitrogen, and stored at −80 °C. Protein concentration and purity were measured using a NanoDrop LITE spectrophotometer (Thermo Scientific).

For ALYREF mutants, MBP column binding buffer (50 mM Tris-HCl 7.4, 5 mM MgCl$_2$·6H$_2$O, 1 M KCl, and 5% glycerol (v/v)) and MBP elution buffer (50 mM Tris-HCl 7.4, 500 mM KCl, 50 mM maltose, and 5% glycerol (v/v)) were used for the purification of the mutants' protein with a truncating region of flexibility. The other purification steps were the same as for the 6×His-MBP-tGFP-ALYREF-WT protein.

## In vitro phase separation assays

Proteins were diluted to buffer with various concentrations of KCl (50 mM Tris-HCl 7.4, X mM KCl, 5% glycerol, 1 mM DTT, 5% PEG4000, X = 50, 75, 150, 300, 400) at room temperature at 25 °C in a total volume of 50 μl. The protein was added as the last component to induce a uniform phase separation. The samples were mixed in microtubules and applied to the bottom of a 35 mm glass confocal petri dish (BS-15-GJM). Images were immediately taken with a Nikon A1R laser confocal scanning microscope and a ×60 oil immersed objective lens (×600). Fluorescent images were analyzed and processed using NIS-Elements Viewer software. Five fields of view were photographed in each condition, and three independent replicates were performed. The fluorescence intensity G value was fixed at 7.81. For experiments with oligonucleotide

fragments with or without m⁵C modification, the protein was fixed at a concentration of 1 μM, and oligonucleotide fragments at different concentrations were added for a total volume of 50 μl. The treatment and fluorescent image method were the same as above. To quantify the effect of RNA oligonucleotide fragments on droplet formation, the sizes of MBP-tGFP-ALYREF droplets were quantified using a counter plugin (Fiji/ImageJ), applying manual thresholds equally to remove background under exactly the same conditions.

## Fluorescence recovery after photobleaching (FRAP) assay

Protein was diluted to 10 μM in 50 mM Tris-HCl 7.4, 50 mM KCl, 5% glycerol, 1 mM DTT, and 5% PEG4000 buffer at room temperature to form droplets. FRAP was performed using an inverted laser confocal scanning microscope (Nikon A1R). Images were acquired with a 60× oil immersion objective, and GFP fluorescence was detected with a 488 nm laser spectral line.

A suitable droplet was selected for magnification (×2400). A circular region with a diameter of approximately 1.5 μm was selected in the region far from the droplet boundary, and bleaching was performed at 488 nm, ~70% of the maximum laser power. The pinhole was set to 49.7 μm. Three images were collected before bleaching, and real-time fluorescence restoration images were collected after bleaching without delay. However, because of the high separation rate, the recovery was actually recorded at a rate of 4 s/frame, for a total of 43 frames. The total fluorescence recovery time was set to 3 minutes. Three independent FRAP experiments were conducted. Fluorescent images were analyzed and processed using NIS-Elements Viewer software. Considering the background and photobleaching effects during the acquisition process, the average fluorescence intensity values of the bleached area (BL) and the background area (BG) were used to calculate the corrected BL (BL_Corr) for each acquisition frame, which was used to estimate the fluorescence intensity of 100%: BL_Corr = BL/BG. An exponential recovery curve was generated using the GraphPad Prism 9 mapping software by plotting the normalized fluorescence intensity value and time.

## Prediction of phase separation characteristics of ALYREF protein

The ALYREF sequence was downloaded from the Uniport website (https://www.uniprot.org/) and imported for analysis on the IUPred2A (IUPred2A (elte.hu)) and Predictor of Natural Disordered Regions website (Predictor of Natural Disordered Regions (PONDR)). The intrinsic disorder regions of the ALYREF protein were predicted. The overall phase separation characteristics of ALYREF protein were predicted by the phase separation protein prediction tool PhaSePred (Main Page-PhaSePred) (Chen et al, 2022).

## Synthesis of oligonucleotide fragments

RNA oligonucleotides with or without (Oligo-m⁵C) modification were synthesized following previously published methods (Yang et al, 2017). The sequence was as follows: 5'-biotin-gagguaugaax-uguaagtt-3' (X = C or m⁵C) (Qingke, China).

## Immunofluorescence assay

Ovarian cancer cells were plated in a glass-bottom cell culture dish (Biosharp, China), washed with PBS, and fixed in 4% paraformaldehyde (Biosharp, China) for 15 min at room temperature. Cells were washed three times with cold PBS for 30 min, permeabilized for 10 min using permeability buffer, and washed with PBS three times (5 min/time). Cells were blocked in blocking buffer for 30 min, washed with PBS three times (5 min/time), and incubated with primary antibodies (at 1:200 dilution) at 4 °C overnight. After three washes in PBS (5 min/time), samples were incubated with secondary antibodies at 1:500 dilution at room temperature for 1 h in the dark. The cells were washed with PBS three times (5 min/time) and incubated with DAPI (Biosharp, China) at room temperature for 10 min. Cells were extensively washed with PBS three times (10 min/time), and images were obtained by laser confocal fluorescence microscopy.

## Data analysis of seq data

For RNA-seq and RIP-seq data, deep sequencing reads were aligned to the human genome (hg38, primary assembly) using STAR version 2.7.8a. The reads were counted using Feature Counts version 2.0.3. The differentially expressed genes were identified following R Bioconductor Package DESeq (Love et al, 2014). For the RNA-BisSeq data, the adapters and low-quality reads from the raw data were filtered using Cutadapt version 1.18. The meRanGs module from meRanTK software version 1.2.0 (Rieder et al, 2016) was selected for reads alignment, and the precisely m⁵C modified bases were distinguished using the meRanCall module. De novo RNA m⁵C motifs were analyzed using HOMER software version 4.11 with the default parameters (http://homer.ucsd.edu/homer). m⁵C profiles in the gene body was drawn by DeepTools version 3.5.0 (Ramírez et al, 2016). For integrative data analysis and statistics, m⁵C sites were defined as the value of m⁵C/C greater than 0.2 and FDR <1e-5. Differentially expressed genes were identified as genes with a $\log_2$(shALY/shNC) >$\log_2$(1.5) and $P$ value < 0.05. Transcripts in which $\log_2$(IP/Input) >$\log_2$(1.5) and $P$ value < 0.05 were determined as ALYREF targets. Overexpression analysis was conducted using the R Bioconductor package Cluster Profiler (Yu et al, 2012) and DAVID (https://david.ncifcrf.gov). Plots were drawn by the R package ggplot2.

## Statistics

Data are presented as mean ± SD from at least three independent experiments. Comparisons between groups were made using Student's $t$ test or one-way ANOVA analysis.

# Data availability

All the high-throughput sequencing raw data have been deposited in the Sequence Read Archive under the accession numbers PRJNA830530 for RNA sequencing, PRJNA830297 for RIP sequencing, and PRJNA830639 for RNA-Bis sequencing. The human cancer data were derived from GEO using the accession codes GSE54388, GSE66957, GSE18520, GSE40595, and GSE10971, the UALCAN database (https://ualcan.path.uab.edu/analysis-prot.html) and the CSIOVDB database (http://csiovdb.mc.ntu.edu.tw/CSIOVDB.html).

Survival analysis data were derived from the Kaplan-Meier Plotter database (https://kmplot.com/analysis/) and the cBioPortal database (http://www.cbioportal.org/). Any additional information required to reanalyse the data reported in this paper is available from the lead contacts upon request.

The source data of this paper are collected in the following database record: biostudies:S-SCDT-10_1038-S44318-025-00657-0.

## Peer review information

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

## Acknowledgements

This research was supported by the Major Program of Hubei Provincial Science Technology Department Fund (2022BCE012), Hubei Provincial Natural Science Innovation and Development Joint Fund (2025AFD214), Innovation and Entrepreneurship Training Program for College Students of Hubei University of

Medicine (S202413249011 and X202510929050), Innovation Research Program for Graduates of Basic Medical College, Hubei University of Medicine (JC2025002). The authors acknowledge the Institutional Animal Care and Use Committee of Hubei University of Medicine and the animal facility for their kind support. We would like to thank Dr. Jia Yu, Yanni Ma, and Yanmin Si (Chinese Academy of Medical Sciences, Beijing) for supporting this study.

## Author contributions

**Hongyan Zhao**: Funding acquisition; Investigation; Writing—original draft. **Qinglv Wei**: Data curation; Investigation; Methodology. **Zhi Luo**: Investigation. **Xiaoyi Liu**: Validation; Investigation. **Chenyue Yang**: Investigation; Methodology. **Ningxuan Chen**: Investigation. **Yuan Wang**: Investigation. **Xin Luo**: Investigation. **Xinzhao Zuo**: Investigation. **Qingya Luo**: Resources. **Yu Yang**: Investigation. **Yang Zhou**: Investigation. **Jiaqi Liu**: Investigation. **Te Zhang**: Investigation. **Dan Yang**: Investigation. **Yingfei Long**: Investigation. **Youchaou Mobet**: Investigation. **Jing Xu**: Resources. **Wei Wang**: Supervision; Project administration. **Tao Liu**: Conceptualization; Supervision; Visualization; Project administration; Writing—review and editing. **Ping Yi**: Conceptualization; Supervision; Project administration.

Source data underlying figure panels in this paper may have individual authorship assigned. Where available, figure panel/source data authorship is listed in the following database record: biostudies:S-SCDT-10_1038-S44318-025-00657-0.

## Disclosure and competing interests statement

The authors declare no competing interests.

# Expanded View Figures

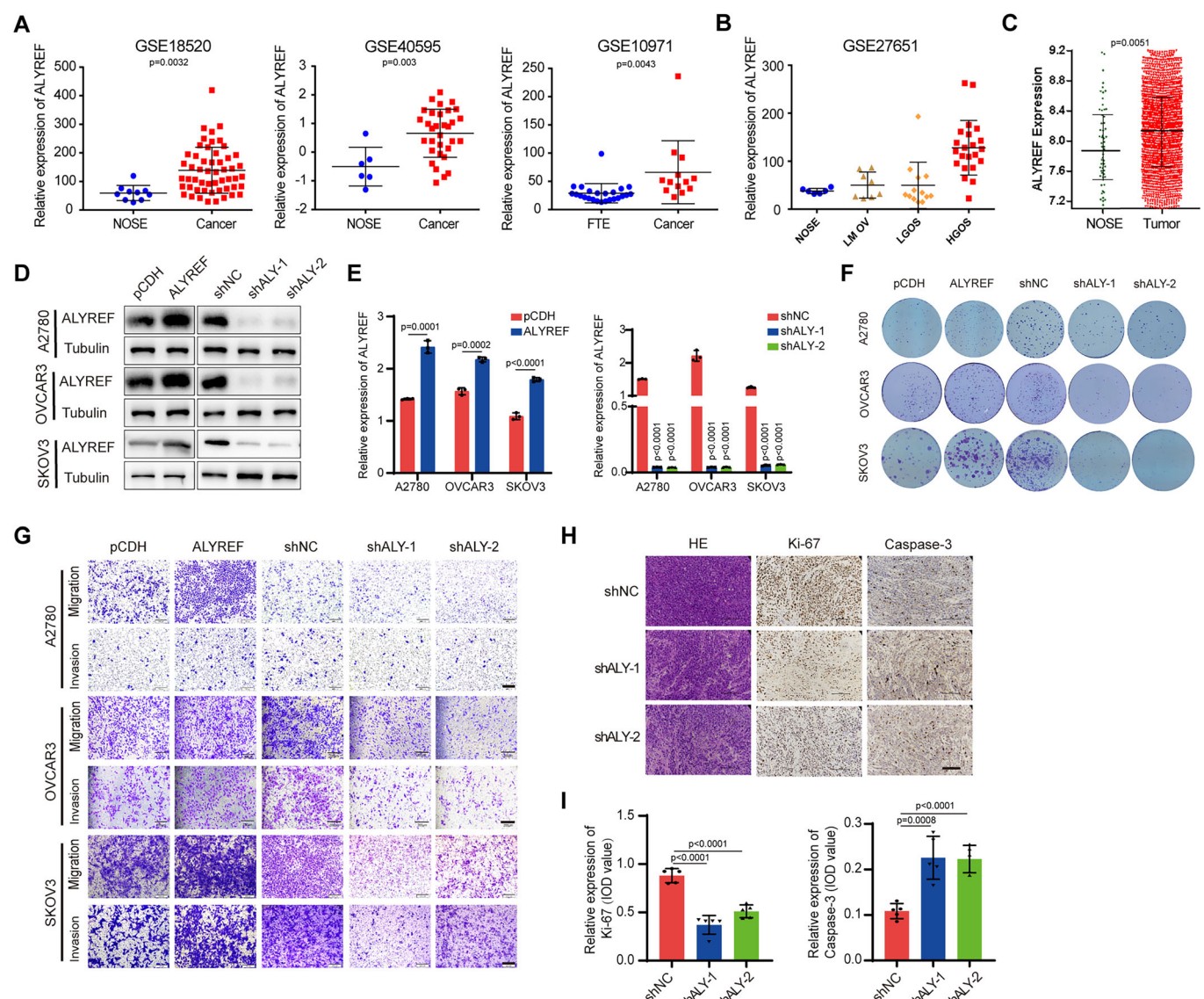

**Figure EV1.  ALYREF is highly expressed in ovarian cancer and promotes tumorigenesis.**

(**A**) Analysis of ALYREF expression in ovarian cancer according to GEO datasets. GSE18520: NOSE ($n = 10$), Cancer ($n = 53$). GSE40595: NOSE ($n = 6$), Cancer ($n = 56$). GSE40595: NOSE ($n = 24$), Cancer ($n = 13$). (**B**) ALYREF was increased in high-grade serous ovarian cancer compared with that in the normal ovarian surface epithelium according to GSE27651. HOSE ($n = 6$),LM OV ($n = 8$), LGOS ($n = 13$), HGOS ($n = 22$). (**C**) ALYREF was overexpressed in ovarian cancer according to the CSIOVDB database. (**D**) Western blotting assays were performed to confirm ALYREF overexpression and knockdown in ovarian cancer cells. (**E**) The expression of the protein was quantified by grayscale in (**D**). $n = 3$ independent experiments. (**F**) The colony formation assays of ovarian cancer cells upon ALYREF overexpression or knockdown. (**G**) The transwell migration and matrigel invasion assays of ovarian cancer cells upon ALYREF overexpression or knockdown. Scale bar, 200 μm. (**H**) The expression of proteins Ki-67 and Caspase-3 was detected in xenografted tumors formed by ovarian cancer cells with ALYREF knockdown through immunohistochemical staining. Scale bar, 100 μm. (**I**) Statistical analysis results of Ki-67 and Caspase-3 positive staining in xenografted tumors ($n = 5$). Data are shown as means ± S.D. *P* value was calculated by one-way ANOVA test with multiple comparisons (**E**, **I**) or unpaired two-sided Student's *t* test (**A**, **C**, **E**). Source data are available online for this figure.

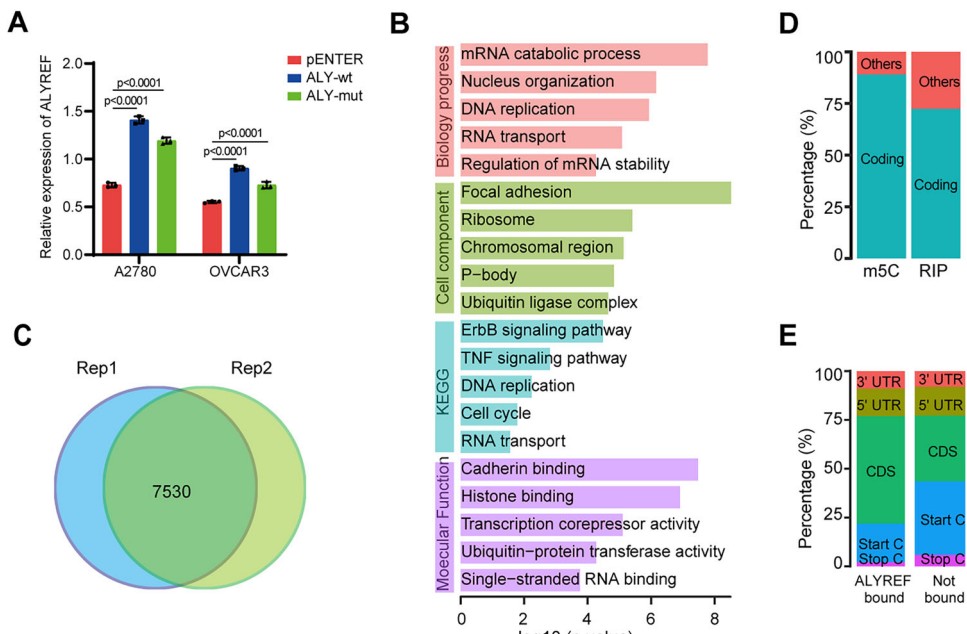

**Figure EV2. ALYREF facilitates ovarian cancer progression in an m⁵C-dependent manner.**

(A) The expression of the protein was quantified by grayscale in ovarian cancer cells with wild-type or mutated ALYREF overexpression. (B) KEGG enrichment analysis revealed that m⁵C-modified transcripts were enriched in signaling pathways such as ErbB, TNF, and cell cycle. (C) RNA-BisSeq analysis showed 7530 transcripts with m⁵C modification. (D) RNA species of ALYREF-binding or m⁵C-modified transcripts. (E) Distribution of m⁵C modification sites on ALYREF-binding transcripts or not-binding transcripts. $n = 3$ independent experiments (A). Data are shown as means ± S.D. $P$ value was calculated by one-way ANOVA test with multiple comparisons (A). Source data are available online for this figure.

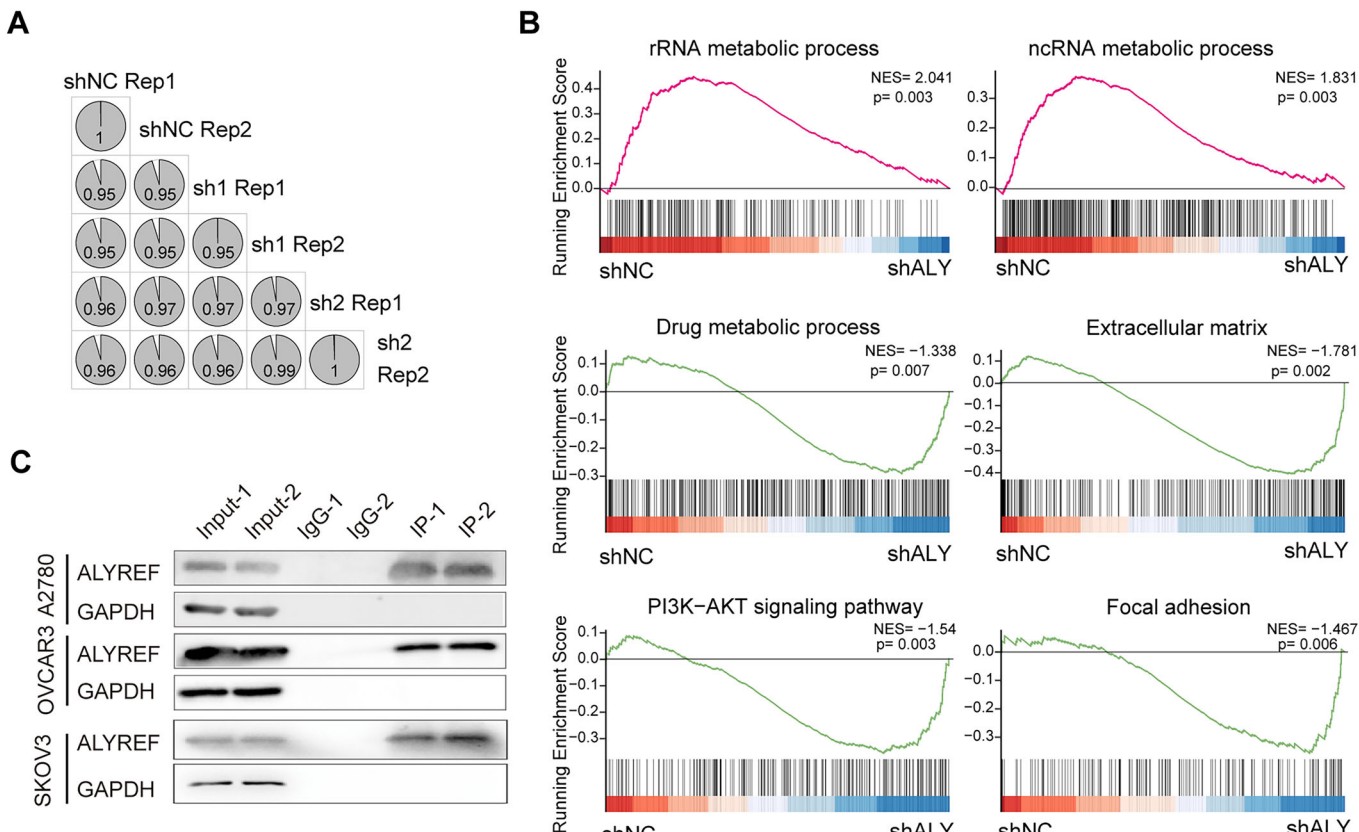

**Figure EV3. Identification of ALYREF-regulated transcripts by RNA-seq, RIP-seq, and RNA-BisSeq.**

(A) Biological repetition analysis of RNA-seq data. (B) The GSEA diagram showed that the functions of genes were downregulated upon ALYREF knockdown. (C) Western blotting confirmed ALYREF immunoprecipitation in RIP assays. *P* value was calculated by permutation test. Source data are available online for this figure.

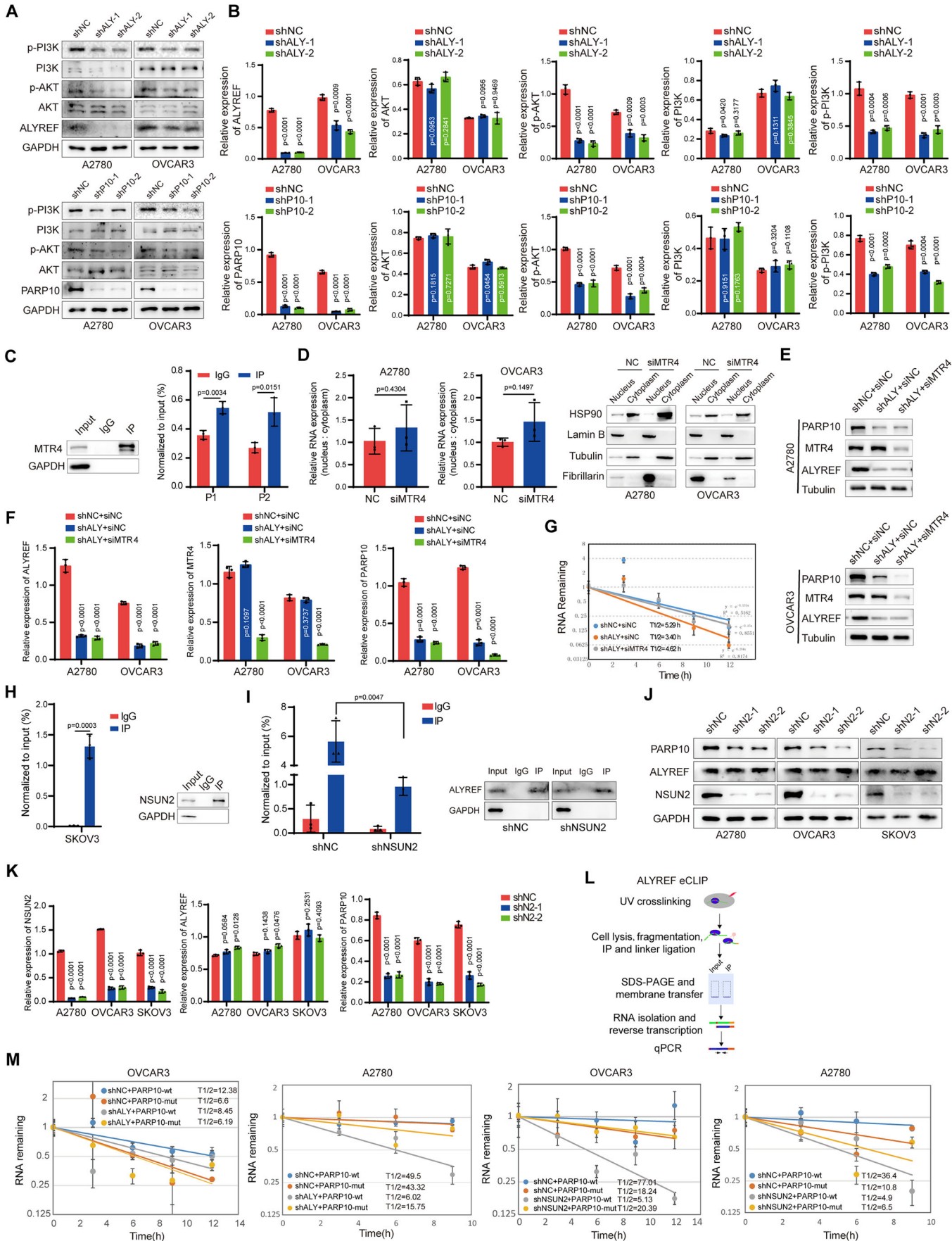

◀ **Figure EV4. ALYREF regulates PARP10 expression in an m⁵C-dependent manner.**

(A) Detecting the protein levels in ovarian cancer cells upon ALYREF knockdown by western blotting. (B) The expression of the protein was quantified by grayscale in (A). (C) RIP assays detecting the interaction between MTR4 and PARP10 mRNA. (D) RT-qPCR detecting the expression level of PARP10 mRNA in nucleocytoplasmic separation assays. (E) Effect of ALYREF knockdown on the protein level of MTR4 in A2780 and OVCAR3 cells. (F) The expression of the protein was quantified by grayscale in (E). (G) ALYREF and MTR4 were knocked down in OVCAR3 cells, and PARP10 mRNA stability was detected. (H) RT-qPCR assays verifying the enrichment of PARP10 in NSUN2 RIP assays in SKOV3 cells. (I) RT-qPCR detecting the enrichment of PARP10 mRNA in ALYREF RIP assays in SKOV3 cells. (J) Detecting protein expression levels in ovarian cancer cells upon NSUN2 knockdown by western blotting assays. (K) The expression of the protein was quantified by grayscale in (J). (L) The diagram of eCLIP experiments of ALYREF in ovarian cancer cells. (M) Effect of ALYREF or NSUN2 knockdown on the stability of wild-type or m⁵C-mutated PARP10 mRNA in ovarian cancer cells. $n = 3$ independent experiments (B–D, F–I, K, M). Data are shown as means ± S.D. *P* value was calculated by one-way ANOVA test with multiple comparisons (B, F, K) or unpaired two-sided Student's *t* test (C, D, H, I). Source data are available online for this figure.

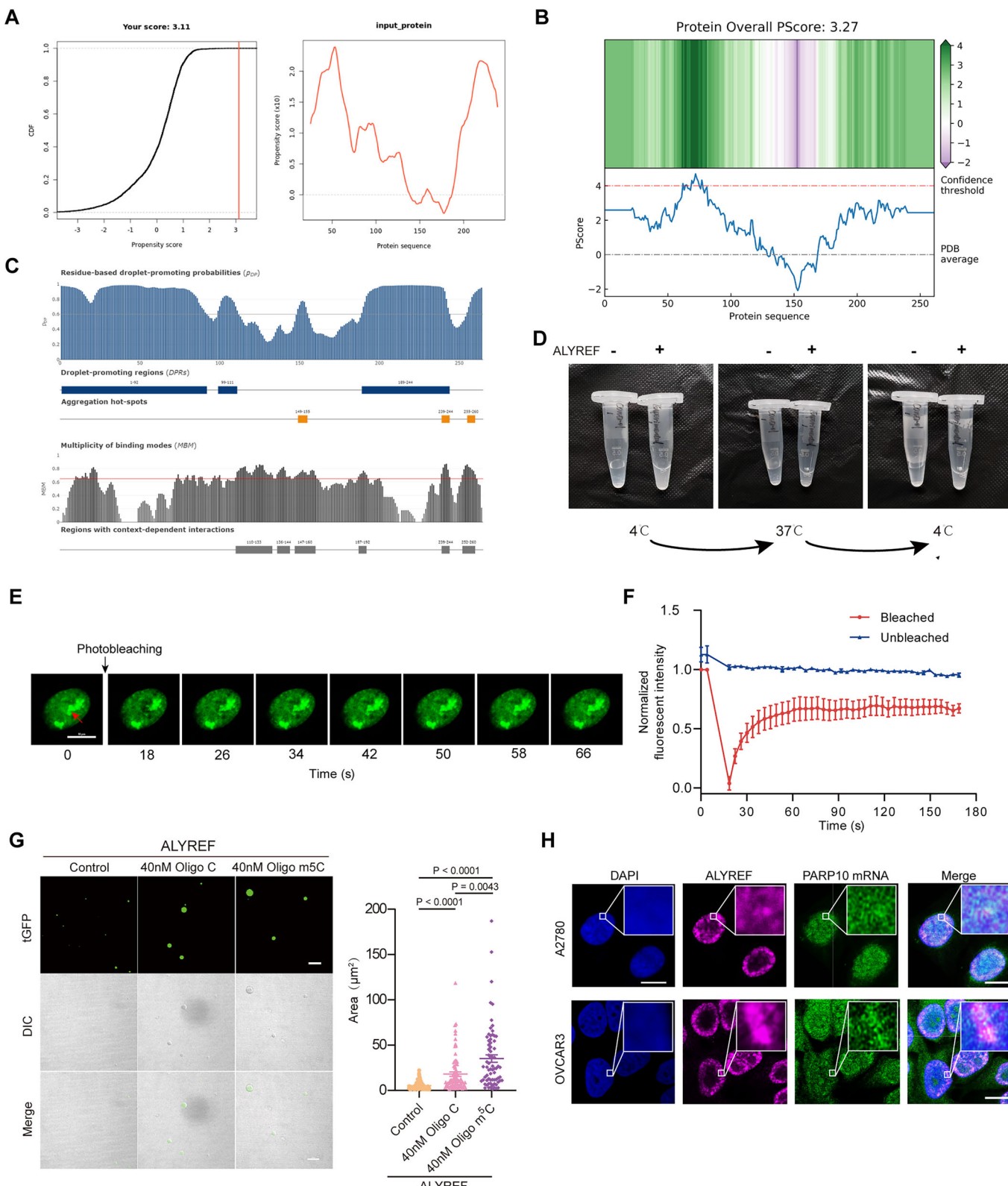

◀ **Figure EV5.  ALYREF undergoes liquid-liquid phase separation.**

(A) Condensation of ALYREF protein was predicted according to catGRANULE. (B) Phase separation of ALYREF was predicted by the π Interaction Score Prediction Tool. (C) The phase separation of ALYREF was predicted by https://fuzdrop.bio.unipd.it/predictor. (D) Effect of temperature on the phase transition properties of ALYREF protein. (E) Fluorescence recovery assays after photobleaching of ALYREF condensates in ovarian cancer cells. The red arrow represents the photobleached region in the droplet. Scale bar, 10 μm. (F) Quantitative analysis of condensates in (E). The black arrow represents the time point of photobleaching. The red arrow represents the photobleached region in the droplet. The red curve represents the average of the normalized fluorescence intensities in different droplet photobleaching regions ($n = 3$ independent experiments). (G) Effect of m$^5$C-modified RNA on phase separation of ALYREF proteins ($n = 50$). Scale bar, 20 μm. (H) Co-localization of ALYREF protein and PARP10 RNA by RNA FISH assays in ovarian cancer cells. Scale bar, 10 μm. Data are shown as means ± S.D. *P* value was calculated by one-way ANOVA test with multiple comparisons (G). Source data are available online for this figure.

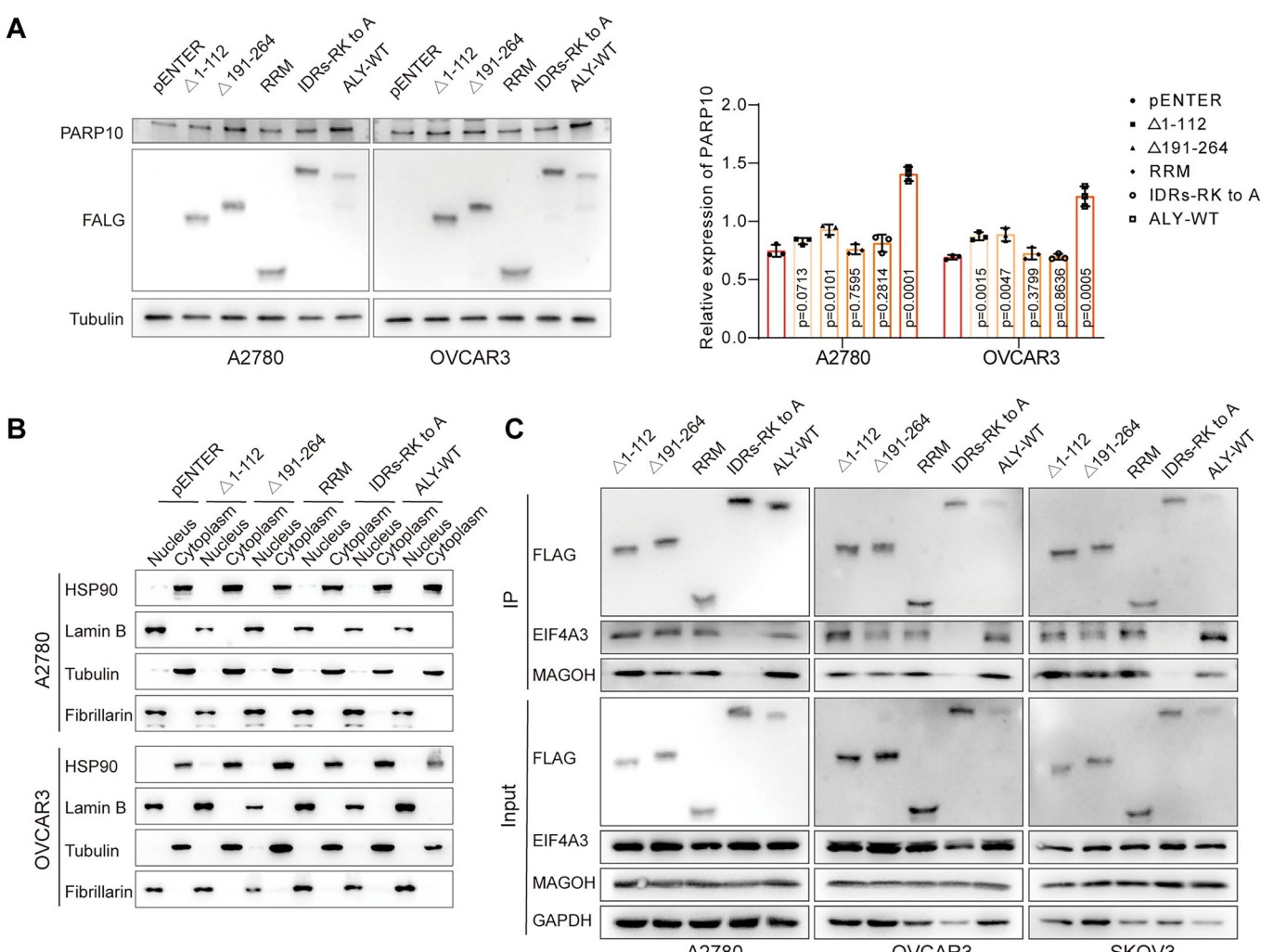

**Figure EV6.  Interaction of ALYREF with EIF3A and MAGOH.**

(**A**) Western blotting verifying the protein of ALYREF-wt, truncations, and mutant expression levels, and quantitative analysis ($n = 3$ independent experiments). Data are shown as means ± S.D. *P* value was calculated by one-way ANOVA test with multiple comparisons. (**B**) Detecting the effectiveness of the nuclear plasma separation assays by western blotting. (**C**) IP assays detecting the interaction of wild-type or mutated ALYREF with EIF3A and MAGOH. Source data are available online for this figure.

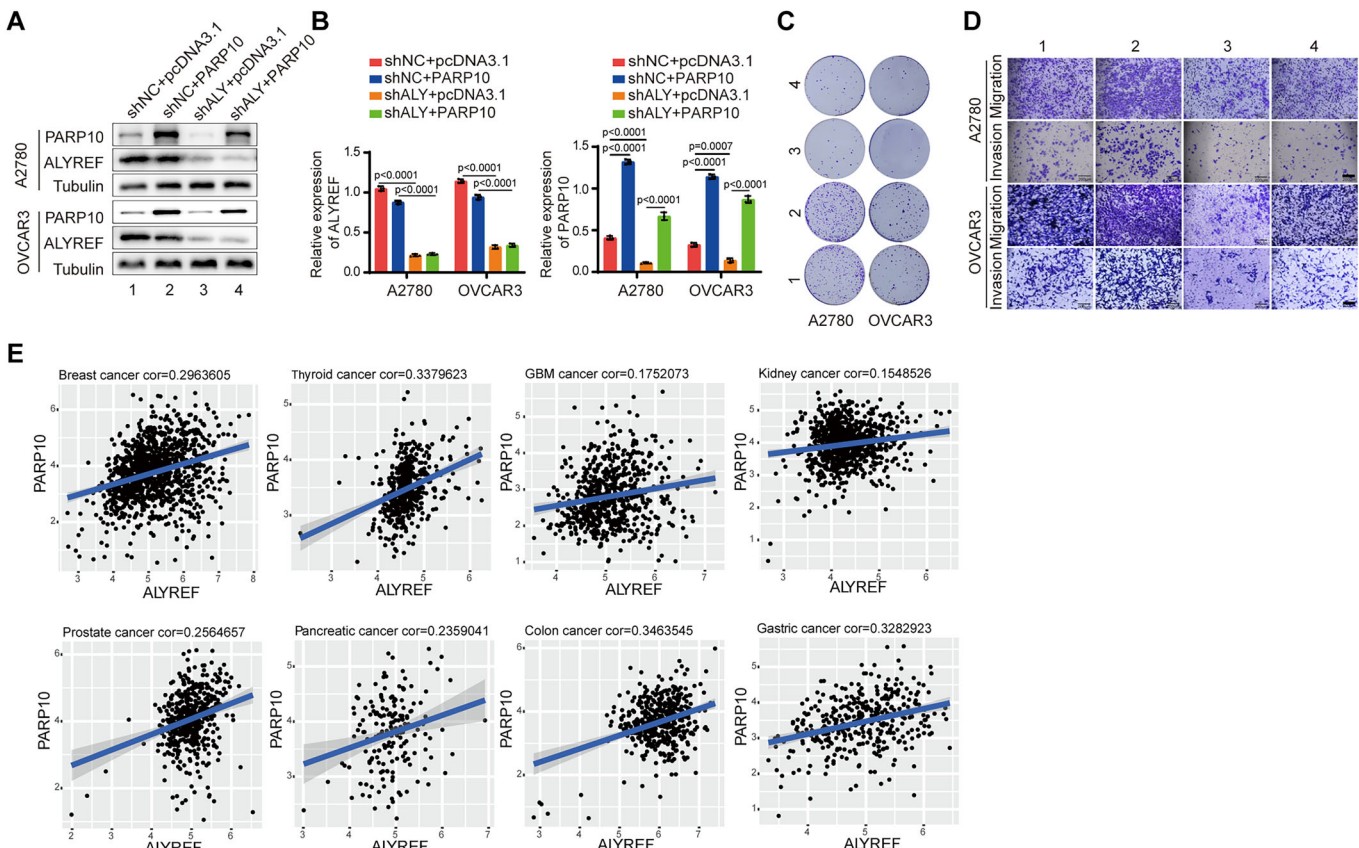

**Figure EV7. Key functions of the ALYREF-PARP10 axis in ovarian cancer.**

(A) Detecting the protein level in ALYREF-depleted ovarian cancer cells with PARP10 overexpression by western blotting. (B) The expression of the protein was quantified by grayscale in (A) ($n = 3$ independent experiments). Data are shown as means ± S.D. *P* value was calculated by one-way ANOVA test with multiple comparisons. (C) Colony formation assays of ALYREF-depleted ovarian cancer cells with overexpression of PARP10. (D) The transwell migration and matrigel invasion assays of ALYREF-depleted ovarian cancer cells with overexpression of PARP10. Scale bar, 100 μm. (E) Correlation analysis between ALYREF and PARP10 expression in various cancers. Source data are available online for this figure.

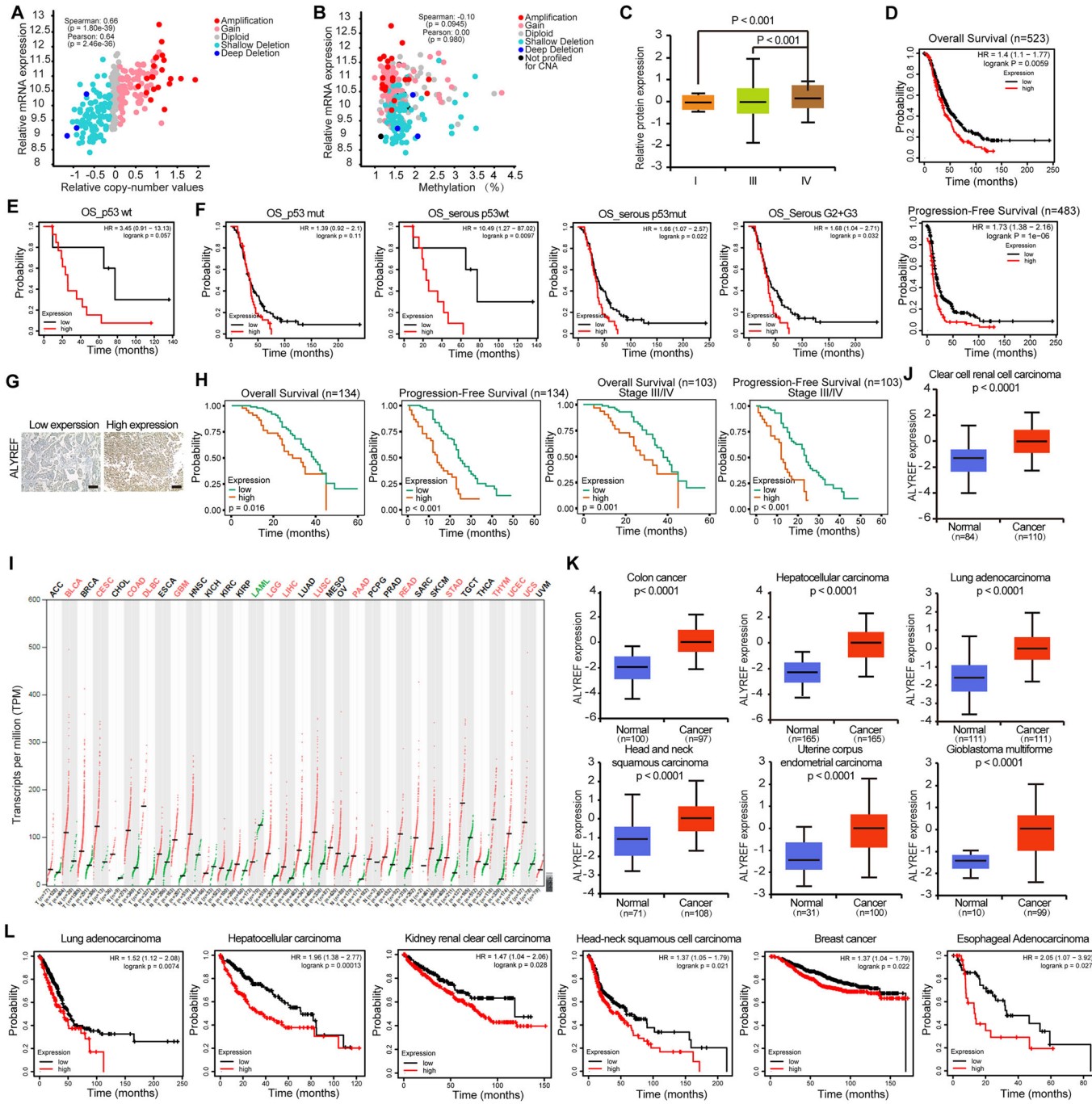

**Figure EV8. ALYREF is highly expressed in pan-cancer.**

(**A**) Correlation analysis of ALYREF mRNA expression with its gene amplification. (**B**) Correlation analysis of ALYREF mRNA expression with DNA methylation. (**C**) ALYREF expression in serous ovarian cancer was significantly correlated with tumor FIGO stage. (**D**) Survival analysis of ovarian cancer patients with different PARP10 expressions according to the Kaplan-Meier Plotter. (**E, F**) Survival analysis of subtype ovarian cancer patients with different ALYREF expressions according to the Kaplan-Meier Plotter. (**G**) Representative IHC images of ALYREF expression in ovarian cancer tissues. Scale bar, 100 μm. (**H**) Survival analysis of ovarian cancer patients with different ALYREF expressions or patients at advanced stage (III and IV) with different ALYREF expressions. (**I–K**) Pan-cancer analysis of ALYREF RNA and protein expression in various cancers according to the TCGA (**I**) and CPTAC databases (**J, K**). (**L**) Survival analysis of cancer patients with different ALYREF expressions according to the Kaplan-Meier Plotter. The central line within the box represents the median value (**C, J, K**). The upper and lower edges of the box represent the 75th and 25th percentiles, respectively. The whiskers extend from the box to the maximum and minimum values within 1.5 times the interquartile range, with any data points beyond this range considered as outliers. *P* value was calculated by two-sided Log-rank (Mantel-Cox) test (**D–F, H, L**) or unpaired two-sided Student's *t* test (**J, K**). Source data are available online for this figure.

