## [Peer Review File · The EMBO Journal]

ALYREF condensation stabilizes m5C-modified PARP10 mRNA and promotes PI3K-AKT signaling in ovarian cancer

Hongyan Zhao, Qinglv Wei, Zhi Luo, Xiaoyi Liu, Chenyue Yang, Ningxuan Chen, Yuan Wang, Xin Luo, Xinzhao Zuo, Qingya Luo, Yu Yang, Yang Zhou, Jiaqi Liu, Te Zhang, Dan Yang, Yingfei Long, Youchaou Mobet, Jing Xu, Wei Wang, Tao Liu, and Ping Yi

Corresponding authors: Ping Yi (yiping@cqmu.edu.cn) , Tao Liu (taoliu@hospital.cqmu.edu.cn), Wei Wang (wangwei@cqmu.edu.cn)

Review Timeline:

Submission Date:	27th Feb 25
Editorial Decision:	26th May 25
Editor's Correspondence:	7th Jun 25
Revision Received:	14th Oct 25
Accepted:	7th Nov 25

Editor: Daniel Klimmeck

Transaction Report:

Please note that the manuscript was previously reviewed at another journal. As EMBO Press has a transfer agreement with that journal, revision was invited based on the reports from that previous external submission.

Point-by-Point Responses to the Comments from Reviewers

Reviewer #1 (Remarks to the Author):

The manuscript entitled “ALYREF forms nuclear phase condensates and stabilizes m⁵C-modified PARP10 mRNA to promote ovarian cancer progression”, **is well written and experiments were performed in a logical order**, although several steps were not really explained. For instance, why was the focus on PARP10 and not on the other 45 target genes of ALYREF. The authors have overexpressed, knocked-down and mutated ALYREF to study its function and the liquid-liquid phase separation (LLPS). They describe that ALYREF binds to m⁵C modified RNA, including PARP10 mRNA. Their findings suggest that PARP10 mRNA stability is enhanced in an m⁵C-dependent manner, which may involve enhanced LLPS of ALYREF. Expression levels of ALYREF seem to be important to prevent binding of MTR4 to PARP10 mRNA. However, this is not further investigated. Furthermore, PARP10 protein levels in contrast to mRNA levels are not restored when both ALYREF and MTR4 are downregulated (Figure 4G/H and Suppl figure 4B). The function of PARP10 is not clear to me. The authors demonstrated that GSK3beta kinase activity goes up in ALYREF or PARP10 downregulated cells. Does this imply that GSK3beta has a tumor suppressor function in ovarian cancer cell lines?

Our reply: We are grateful for the reviewer's acknowledgment of our work and constructive comments.

We apologize for the unclear rationale for selecting PARP10 as the target of ALYREF in the primary manuscript. We are also sorry for the mistake made in the primary manuscript. Indeed, based on the results of RNA-seq, RIP-seq, and m⁵C-seq, we obtained 26 but not 46 target genes of ALYREF (Fig. 3C in the revised manuscript). Among these 26 potential targets, 7 genes' expression was downregulated and 19 genes' expression was upregulated upon ALYREF knockdown according to RNA-seq results. Notably, the cumulative curve analysis showed that m⁵C-modified transcripts

tended to be decreased upon ALYREF knockdown, and ALYREF-bound transcripts were also tended to be declined upon ALYREF knockdown, suggesting that ALYREF binds to m⁵C-modified transcripts and promotes their expression (Fig. 3D, E in the revised manuscript). Thus, we selected these 7 downregulated genes for further experimental verification. RT-qPCR confirmed downregulation of these genes in ovarian cancer cells upon ALYREF knockdown (Fig. 3I-K in the revised manuscript). RIP assays showed that target mRNAs were significantly enriched by ALYREF (Fig. 3L-N and Fig. EV3C in the revised manuscript). Functional enrichment analysis of the differentially expressed genes upon ALYREF knockdown in RNA-seq showed that ALYREF was involved in the PI3K-AKT signaling pathway (Fig. 3B in the revised manuscript). PARP10 has been demonstrated to be involved in cancer progression via regulation of PI3K-AKT signaling pathways. The PARP10 expression fold-change in the RNA-seq results ranks first. Our RT-qPCR results also showed that PARP10 was significantly downregulated upon ALYREF depletion in ovarian cancer cells (Fig. 3I-N in the revised manuscript). The PARP/ARTD family includes 17 protein-modifying enzymes that share a conserved catalytic domain containing ADP-ribosylation activity. PARP10 promotes cell proliferation and tumorigenesis in cervical cancer cells by alleviating replication stress^[1]. Moreover, previous analysis of the TCGA database revealed that PARP10 expression is increased in 19% of breast tumors, and its overexpression facilitates cell proliferation and tumor progression^[2]. Thus, we speculated that PARP10 might be the key target of ALYREF in ovarian cancer.

In our study, the results showed that the stability of PARP10 mRNA was reduced upon ALYREF knockdown. The exosome complex is responsible for the processing and degradation of nuclear RNAs, and MTR4 is an important part of the exosome complex. RIP assays showed that MTR4 did not bind PARP10 mRNA in ALYREF-expressing ovarian cancer cells (Fig. EV4C in the revised manuscript). However, the interaction between MTR4 and PARP10 mRNA was significantly increased upon ALYREF knockdown (Fig. 4F in the revised manuscript). ALYREF

deficiency did not affect MTR4 expression in ovarian cancer cells (Fig. EV4E, F in the revised manuscript). These results support a model by which ALYREF and MTR4 competitively bind to PARP10 mRNA to regulate its expression. To confirm this possibility, we knocked down MTR4 in ALYREF-depleted ovarian cancer cells. MTR4 knockdown restored PARP10 mRNA but not protein expression in ALYREF-depleted ovarian cancer cells (Fig. 4G and Fig. 4E, F in the revised manuscript). Furthermore, ALYREF knockdown decreased the stability of PARP10 mRNA, while concomitant knockdown of MTR4 promoted PARP10 mRNA stability (Fig. 4H and Fig. EV4G in the revised manuscript). Thus, ALYREF deficiency suppressed the nuclear export of PARP10 mRNA to the cytoplasm for translation, which resulted in that PARP10 protein was not recovered upon MTR4 knockdown in ALYREF-deficient cells through PARP10 mRNA detained in the nucleus was recovered upon MTR4 knockdown in ALYREF-deficient cells.

Also, through a series of LLPS *in vitro* and *in vivo* (Fig. 5 and Fig. EV5 in the revised manuscript), we demonstrated that ALYREF undergoes LLPS. More importantly, LLPS depletion abrogated ALYREF's functions on the expression as well as the stability of PARP10 mRNA (Fig. 6 in the revised manuscript).

PARP10 plays different roles in different tumors. PARP10 belongs to the PARP/ARTD family, which includes 17 protein-modifying enzymes that share a conserved catalytic domain containing ADP-ribosylation activity. PARP10 catalyzes the transfer of a single ADP moiety to target proteins and affects protein functions. PARP10 promotes cell proliferation and tumorigenesis in cervical cancer cells by alleviating replication stress^[1]. Moreover, previous analysis of the TCGA database revealed that PARP10 expression is increased in 19% of breast tumors, and its overexpression facilitates cell proliferation and tumor progression^[2]. However, PARP10 acts as a tumor suppressor gene in hepatocellular carcinoma^[3], suggesting the complex role of PARP10. It has been reported that PARP10 promotes the mono-ADP-ribosylation of GSK3 β and inhibits its kinase activity in the progression

of pathological hypertrophy. In our study, we revealed that ALYREF promoted ovarian cancer progression through upregulating PARP10 expression, and overexpression of PARP10 substantially mitigated the tumor-suppressing effect of ALYREF depletion on ovarian cancer. Thus, we detected the kinase activity of GSK3 β upon ALYREF knockdown or PARP10 knockdown and found that either ALYREF knockdown or PARP10 knockdown increased the kinase activity of GSK3 β in the primary manuscript. The role of GSK3 β in cancer is also complex with oncogenes or tumor suppressors in different cancers, and targeting GSK3 β can suppress or promote cancer progression in the context of different cancers. As a tumor suppressor, GSK3 β mainly targets β -catenin and induces its phosphorylation and degradation, thus inhibiting Wnt signaling and tumor growth. However, we detected the expression of β -catenin in ovarian cancer cells upon ALYREF knockdown and found no significant changes in β -catenin expression (data not shown), suggesting that GSK3 β might not be the key downstream target of ALYREF and PARP10 in ovarian cancer progression. Thus, we have reanalyzed the function enrichment of genes regulated by ALYREF according to RNA-seq results and found that PI3K-AKT signaling pathway was significantly enriched. PARP10 has also been reported to affect the PI3K-AKT signaling pathway^[4]. We speculated that ALYREF might regulate PI3K-AKT signaling through PARP10. As expected, either ALYREF knockdown or PARP10 knockdown significantly decreased the phosphorylation levels of PI3K and AKT in ovarian cancer cells (Fig. EV4A, B in the revised manuscript). Accordingly, we have revised the manuscript.

Major comments:

1. The authors use 3 ovarian cancer cell lines. They reflect different subtypes: A2780 is an endometrioid cell line with wild type p53. OVCAR3 is a high grade serous ovarian cancer cell line with mutant p53. SKOV3 is a p53-null cell line with mutant ARID1A. Results with A2780 seem to give the largest changes when ALYREF or PARP10 knock-downs are used. Can the authors comment on this? Moreover, not all 3 cell lines are used in all experiments, why not? May the different subtypes also have

an effect on their analyses of patient data. In my opinion subanalyses based on the different subtypes should be included as well.

Our reply: We sincerely appreciate the reviewer's valuable comments. In the course of our assays, the largest changes might be due to the fact that the efficiency of virus infection and plasmid transfection in A2780 cells was better than that of the OVCAR3 and SKOV3 cell lines. Following the reviewer's suggestion, we have performed additional experiments in A2780 ovarian cancer cells to validate our characterized mechanisms. Though the three ovarian cancer cell lines used in this study reflect different cancer subtypes, ALYREF exhibited a similar effect on the proliferation and metastasis of these three ovarian cancer cell lines (Fig. 1 and Fig. EV1 in the revised manuscript), and ALYREF regulated PARP10 expression in all three ovarian cancer cells (Fig. 3I-N, Fig. 4 and Fig. EV4 in the revised manuscript), suggesting that ALYREF might function as a tumor-promoting role through PARP10 in ovarian cancer. In addition, we have performed some subanalysis of associations between ALYREF expression and the prognosis of ovarian cancer patients with the different subtypes. The results showed that higher expression of ALYREF portended a worse prognosis for ovarian cancer patients regardless of p53 status or tumor grades (Fig. EV8D, E in the revised manuscript). Specifically, for serous ovarian cancer patients with either wild-type p53 or mutated p53, higher expression of ALYREF predicted shorter overall survivals. Similar results were also observed in high grade serous ovarian cancer patients. Despite no significant differences due to limited patients' clinical data, either wild-type p53 or p53 mutated patients with higher ALYREF expression tended to undergo shorter overall survivals (Fig. EV8D, E in the revised manuscript). Also, we found an inconsistent but unconvincing result of associations between ALYREF expression and endometrioid ovarian cancer patients owing to the substantially limited patient number (data not shown). Thus, subanalyses of associations between ALYREF expression and cancer patients' prognosis based on the different subtypes should be analyzed in further investigations, though higher ALYREF expression predicted poor prognosis for a variety of cancers.

2. The authors focus on PARP10, why not on the other 45 genes? Was a gene set enrichment performed for these 46 genes? Why was PARP10 selected and was MYC part of this set?

Our reply: We apologize for the unclear rationale for selecting PARP10 as the target of ALYREF in the primary manuscript, as mentioned above. We are also sorry for the mistake made in the primary manuscript. As mentioned above, indeed, based on the results of RNA-seq, RIP-seq, and m⁵C-seq, we obtained 26 but not 46 target genes of ALYREF (Fig. 3C in the revised manuscript). Among these 26 potential targets, 7 genes' expression was downregulated and 19 genes' expression was upregulated upon ALYREF knockdown according to RNA-seq results. Notably, the cumulative curve analysis showed that m⁵C-modified transcripts tended to be decreased upon ALYREF knockdown, and ALYREF-bound transcripts were also tended to be declined upon ALYREF knockdown, suggesting that ALYREF binds to m⁵C-modified transcripts and promotes their expression (Fig. 3D, E in the revised manuscript). Thus, we selected these 7 genes for further experimental verification. RT-qPCR confirmed downregulation of these genes in ovarian cancer cells upon ALYREF knockdown (Fig. 3I-K in the revised manuscript). RIP assays showed that target mRNAs were significantly enriched by ALYREF (Fig. 3L-N and Fig. EV3C in the revised manuscript). Functional enrichment analysis of the differentially expressed genes upon ALYREF knockdown in RNA-seq showed that ALYREF was involved in the PI3K-AKT signaling pathway (Fig. 3B in the revised manuscript). PARP10 has been demonstrated to be involved in cancer progression via regulation of PI3K-AKT signaling pathways. The PARP10 expression fold-change in the RNA-seq results ranks first. Our RT-qPCR results also showed that PARP10 was significantly downregulated upon ALYREF depletion in ovarian cancer cells (Fig. 3I-N in the revised manuscript). The PARP/ARTD family includes 17 protein-modifying enzymes that share a conserved catalytic domain containing ADP-ribosylation activity. PARP10 promotes cell proliferation and tumorigenesis in cervical cancer cells by alleviating replication stress^[1]. Moreover, previous analysis of the TCGA database

revealed that PARP10 expression is increased in 19% of breast tumors, and its overexpression facilitates cell proliferation and tumor progression^[2]. Thus, we speculated that PARP10 might be the key target of ALYREF in ovarian cancer. We also performed a gene set enrichment of 26 potential targets, and only three terms including DNA metabolic process, embryonic organ development, and chromatin organization were enriched (data not shown). Though PARP10 was demonstrated to be involved in DNA repair^[1], the underlying mechanism was largely unknown. We showed that ALYREF regulated PI3K-AKT signaling through PARP10 in ovarian cancer cells to promote ovarian cancer progression, and thus the gene set enrichment results were not included in the revised manuscript. In addition, MYC was not identified in our integrated multi-omics analysis as the target of ALYREF in ovarian cancer cells.

3. Can the authors explain why PARP10 protein levels in contrast to mRNA levels are not restored when both ALYREF and MTR4 are downregulated (Figure 4G/H and Suppl figure 4B)? Why is MTR4 only playing a role in the absence of ALYREF and is this cell line dependent? Is MTR4 affecting PARP10 mRNA localization in the nucleus?

Our reply: We sincerely appreciate the reviewer's valuable comments. In our study, the results showed that the stability of PARP10 mRNA was reduced upon ALYREF knockdown. The exosome complex is responsible for the processing and degradation of nuclear RNAs, and MTR4 is an important part of the exosome complex. RIP assays showed that MTR4 did not bind PARP10 mRNA in ALYREF-expressing ovarian cancer cells (Fig. EV4C in the revised manuscript). However, the interaction between MTR4 and PARP10 mRNA was significantly increased upon ALYREF knockdown (Fig. 4F in the revised manuscript). ALYREF deficiency did not affect MTR4 expression in ovarian cancer cells (Fig. EV4E, F in the revised manuscript). These results support a model by which ALYREF and MTR4 competitively bind to PARP10 mRNA to regulate its expression. To confirm this possibility, we knocked down MTR4 in ALYREF-depleted ovarian cancer cells. MTR4 knockdown restored

PARP10 mRNA but not protein expression in ALYREF-depleted ovarian cancer cells (Fig. 4G and Fig. EV4E, F in the revised manuscript). Furthermore, ALYREF knockdown decreased the stability of PARP10 mRNA, while concomitant knockdown of MTR4 promoted PARP10 mRNA stability (Fig. 4H and Fig. EV4G in the revised manuscript). ALYREF deficiency suppressed the nuclear export of PARP10 RNA to the cytoplasm for translation, which resulted in the fact that PARP10 protein did not recover upon MTR4 knockdown in ALYREF-deficient cells through PARP10 mRNA detained in the nucleus was recovered upon MTR4 knockdown in ALYREF-deficient cells. We also detected the PARP10 mRNA localization upon MTR4 knockdown in ALYREF-proficient cells and found that MTR4 didn't affect the PARP10 localization in the nucleus (Fig. EV4D in the revised manuscript).

4. Is PARP10 expression related to ALYREF expression in other tumor types?

Our reply: We have performed correlation analysis of PARP10 and ALYREF expression in a variety of tumor types and found that PARP10 expression was positively correlated with ALYREF expression in multiple cancers (Fig. EV7E in the revised manuscript).

5. What is the role of GSK3beta, these results should be discussed.

Our reply: We sincerely appreciate the reviewer's valuable comments. As mentioned above, the role of GSK3 β in cancer is also complex with oncogenes or tumor suppressors in different cancers, and targeting GSK3 β can suppress or promote cancer progression in the context of different cancers. As a tumor suppressor, GSK3 β mainly targets β -catenin and induces its phosphorylation and degradation, thus inhibiting Wnt signaling and tumor growth. However, we detected the expression of β -catenin in ovarian cancer cells upon ALYREF knockdown and found no significant changes in β -catenin expression (data not shown), suggesting that GSK3 β might not be the key downstream target of ALYREF and PARP10 in ovarian cancer progression. Thus, we have reanalyzed the function enrichment of genes regulated by ALYREF according to RNA-seq results and found that PI3K-AKT was enriched. PARP10 has also been

reported to affect the PI3K-AKT signaling pathway^[4]. We speculated that ALYREF might regulate PI3K-AKT signaling through PARP10. As expected, either ALYREF knockdown or PARP10 knockdown significantly decreased the phosphorylation levels of PI3K and AKT in ovarian cancer cells (Fig. EV4A, B in the revised manuscript). Accordingly, we have revised the manuscript.

Grammars and typos:

- Please add the genename when mRNA levels are indicated on the Y-axis.
- Legend of Extended data Fig 7D is missing.
- which exist intrinsic disordered regions play key roles in promoting tumorigenesis and development something missing?
- Line 41 m6A = m⁶A
- Line 310 m5C= m⁵C
- Line 319-320 modifaciton= modification

Our reply: We have revised the manuscript as required including adding the gene name on the Y-axis and adding a legend of in Fig. EV7D in the revised manuscript. We have also performed a series of cell phenotype assays *in vitro* and *in vivo* to confirm the effect of LLPS on ALYREF promoting ovarian cancer. Truncations, mutants, and wild-type ALYREF were overexpressed in ovarian cancer cells, respectively. ALYREF overexpression promoted the proliferation, migration, and invasion abilities of ovarian cancer cells (Fig. 7A-C in the revised manuscript). However, LLPS depletion of ALYREF failed to facilitate ovarian cancer cell proliferation, migration, and metastasis (Fig. 7A-C in the revised manuscript). Consistently, *in vivo* assays also showed that LLPS depletion abrogated the promoting effect of ALYREF on tumor growth in mice (Fig. 7D-F in the revised manuscript).

Reviewer #2 (Remarks to the Author):

This article reports that the m⁵C binding and mRNA export protein, ALYREF, is expressed at elevated levels in ovarian cancer cells, inducing a hyperproliferative phenotype. Using a variety of established ovarian cancer cell lines, the authors show

that knocking down ALYREF reduces colony formation, and cell migration and invasion. Further, in mouse models, they show that tumor volume and number are reduced with ALYREF knockdown. The authors next showed that ALYREF's hyperproliferative effects are dependent on its binding to m⁵C and that it binds the 5' end of the coding regions of large numbers of mRNAs. By overlaying the results of RNA-seq, RIP-seq and RNA-bisseq experiments, the authors identified 46 mRNAs that are most highly influenced by ALYREF knockdown. They used RT-qPCR to confirm the effects of ALYREF knockdown on mRNA levels for ~8 specific genes and further showed that the ALYREF protein is bound to different extents to these mRNAs. The authors focused attention on ALYREF-dependent regulation of PARP10 expression, showing that ALYREF binds PARP10 mRNA and enhances its expression. The article next makes a detour to discuss phase separation by ALYREF, both in vitro in purified form and in cells, with the contention being that phase separation is related to the function of ALYREF. The closing sections of the article address functional interplay between ALYREF and PARP10, with the results showing that ALYREF expression enhances the hyperproliferative phenotype associated with PARP10 overexpression in ovarian cancer cell lines. Finally, the authors performed a pan-cancer analysis, showing that ALYREF gene is amplified at low frequencies in many cancer types. Also, the authors show that ovarian cancer patients with high ALYREF expression have slightly poorer outcomes than those with low ALYREF expression.

Overall, the data presented in the manuscript appear to be of high quality and true. Generally, the authors' statements and conclusions are supported by data, with an exception being conclusions drawn from data related to phase separation by ALYREF. ALYREF has been reported previously to be expressed at elevated levels in other cancer types; the demonstration of this for ovarian cancer represents an incremental advance. The link between ALYREF and PARP10 is interesting, but, apart from establishing synergy between expression of the two proteins and enhanced cell proliferation, the manuscript does not provide new mechanistic insights. For

example, how does ALYREF-dependent enhanced PARP10 expression promote the hyperproliferative phenotype of ovarian cancer cells. The section of the manuscript on phase separation by ALYREF is underdeveloped and adds little insight into the mechanism(s) underlying ALYREF-dependent mRNA regulation. Detailed comments are provided below. For these reasons, the manuscript fails to provide the level of novelty and impact expected for publication in [other journal].

Our reply: We are grateful for the reviewer's acknowledgment of our work. In this study, we first showed that ALYREF forms condensates in the nucleus by liquid-liquid phase separation and promotes ovarian cancer malignancy by increasing the proliferation and metastasis of ovarian cancer cells. ALYREF binds to m⁵C-modified PARP10 mRNA competitively with MTR4, thereby enhancing the stability and nuclear export of PARP10 mRNA and activating downstream PI3K-AKT signaling. LLPS is indispensable for ALYREF promoting PARP10 expression and mRNA stability as well as ovarian tumorigenesis. Furthermore, ALYREF is highly expressed in cancers including ovarian cancer, and predicts a poor prognosis for cancer patients.

As suggested, we have performed a series of additional experiments to enhance the conclusions of ALYREF phase separation as well as its links with PARP10 regulation. Besides *in vitro* experiments, we constructed stable ovarian cancer cells overexpressed with ALYREF labeled with tGFP and conducted FRAP *in vivo*. The results showed that the recovery time of the OVCAR3 cell line and the A2780 cell line was about 105 s and 66 s, respectively (Fig. 5I, J, and Fig. EV5E, F in the revised manuscript). RNA fluorescence *in situ* hybridization assay showed that PARP10 mRNA was localized in ALYREF condensates in the nucleus of ovarian cancer cells (Fig. EV5H in the revised manuscript). To map the key domains for its phase separation, we constructed the mutants of ALYREF. *In vitro* phase separation assays demonstrated that the N-terminal IDR and C-terminal IDR together drove LLPS of ALYREF, and ALYREF phase separation disappeared when the IDRs were absent. Moreover, mutations of Arg and Lys in IDRs to Ala also decreased ALYREF

condensation (Fig. 6B in the revised manuscript). The aggregates were significantly smaller, and PARP10 upregulation was disrupted in cells expressing the ALYREF mutants compared with cells with wild-type ALYREF expression (Fig. 6C, D in the revised manuscript). To study the regions where ALYREF performs phase separation, the additional opto-droplet system was constructed, and photo induced IDR phase separation assays were carried out. The results show that the phase separation phenomenon of ALYREF occurs in the case of full-length ALYREF (Fig. 6E, F in the revised manuscript).

To examine whether LLPS was involved in ALYREF regulation of PARP10 expression, we constructed GFP-ALYREF and its respective truncated forms in which IDRs were depleted or mutated and transfected the constructs into ovarian cancer cells. RT-qPCR showed that unlike wild-type ALYREF, which promoted PARP10 expression, all the mutated forms of ALYREF exerted little effect on the level of PARP10 mRNA (Fig. 6G and Fig. EV6A in the revised manuscript). The nucleoplasmic isolation and RNA stability assays also showed that the mutated forms of ALYREF did not affect the nucleoplasmic distribution and stability of PARP10 mRNA (Fig. 6H, I, and Fig. EV6B in the revised manuscript). To detect whether ALYREF mutants with LLPS loss affected its interaction with other components in the TREX complex, we performed immunoprecipitation assays and found that only mutation of R/K to A destructed the interaction of ALYREF with EIF4A3 and MAGOH, whereas other LLPS-depleted ALYREF mutants were intactly bound to EIF4A3 and MAGOH. ALYREF interacts with other proteins primarily through amino acid sequences in the 55-182 region^[5]. These results suggest that LLPS depletion interferes with ALYREF-regulating PARP10 expression and is independent of its interaction with other components in the TREX (Fig. EV6C in the revised manuscript).

Additionally, to confirm the effect of LLPS on ALYREF promoting ovarian cancer, truncations, mutants, and ALY-wt were overexpressed in ovarian cancer cells,

respectively. ALY-wt overexpression promoted the proliferation, migration, and invasion abilities of ovarian cancer cells (Fig. 7A-C in the revised manuscript). However, LLPS depletion of ALYREF failed to facilitate ovarian cancer cell proliferation, migration, and metastasis (Fig. 7A-C in the revised manuscript). Consistently, *in vivo* assays also showed that LLPS depletion abrogated the promoting effect of ALYREF on tumor growth in mice (Fig. 7D-F in the revised manuscript). Together, these results demonstrate that ALYREF promotes PARP10 expression and tumorigenesis dependently on LLPS.

Major concern

1. The results in Fig. 5 showing that ALYREF forms what appear to be phase separated condensates *in vitro* are interesting but, without further experimentation, they cannot be related to the function of ALYREF in cells. The authors used IF in fixed cells to characterize the localization patterns of ALYREF and several mutants, but these data fall far short of establishing that ALYREF forms phase separated condensates in cells. Live cell imaging with fluorescently tagged proteins would be needed to begin to address this issue.

Our reply: We are grateful for the reviewer's comments. As suggested, to confirm phase-separated condensation of ALYREF in cells, we have performed a series of additional experiments. We constructed stable ovarian cancer cells overexpressed with ALYREF labeled with tGFP and conducted FRAP *in vivo*. The results showed that the recovery time of the OVCAR3 cell line and the A2780 cell line was about 105 s and 66 s, respectively (Fig. 5I, J, and Fig. EV5E, F in the revised manuscript). ALYREF contains three major domains: an N-terminal arginine-lysine-rich (R/K) domain, a central RNA-recognition motif domain (RRM), and a C-terminal arginine-lysine-rich (R/K) domain. ALYREF was predicted to contain IDRs in both the N-terminal and C-terminal domains (Fig. 6A in the revised manuscript). To map the key domains for its phase separation, we constructed the mutants of ALYREF. *In vitro* phase separation assays demonstrated that the N-terminal IDR and C-terminal IDR together drove LLPS of ALYREF, and ALYREF phase separation disappeared

when the IDRs were absent. Moreover, mutations of Arg and Lys in IDRs to Ala also decreased ALYREF condensation (Fig. 6B in the revised manuscript). The aggregates were significantly smaller, and PARP10 upregulation was disrupted in cells expressing the ALYREF mutants compared with cells with wild-type ALYREF expression (Fig. 6C, D in the revised manuscript). In addition, to study the regions where ALYREF performs phase separation, the opto-droplet system was constructed, and photo induced IDR phase separation assays were carried out. The results show that the phase separation phenomenon of ALYREF occurs in the case of full-length ALYREF (Fig. 6E, F in the revised manuscript).

ALYREF is a subunit within the TREX complex that regulates mRNA export from the nucleus (see <https://doi.org/10.1038/s41586-023-05904-0>), but the authors do not consider this mechanistic context. The effects of the ALYREF truncations and mutants the authors studied could be due to disruption of interactions with other components in the TREX complex (e.g., EIF4A3, MAGOH, and Y14) rather than effects on phase separation, as argued by the authors. This alternative interpretation of their data is not considered by the authors.

Our reply: We sincerely appreciate the reviewer's valuable comments. To detect whether ALYREF mutants with LLPS loss affected its interaction with other components in the TREX complex, we performed immunoprecipitation assays and found that only mutation of R/K to A destructed the interaction of ALYREF with EIF4A3 and MAGOH, whereas other LLPS-depleted ALYREF mutants were intactly bound to EIF4A3 and MAGOH. ALYREF interacts with other proteins primarily through amino acid sequences in the 55-182 region^[5]. These results suggest that LLPS depletion interferes with ALYREF-regulating PARP10 expression and is independent of its interaction with other components in the TREX (Fig. 6C in the revised manuscript).

Minor concerns

1. Fig. 3F, G. The statistics of similarities and differences in the results are not

evaluated and need to be. The results in panel F seem very similar to each other, in contrast to the authors' statements in the text. Lines 136-137: "downregulated transcripts bound by ALYREF tended to undergo m⁵C modification in their CDS and 5'-UTR regions (Fig. 3f, g)."

Our reply: We are grateful for the reviewer's comments. As suggested, we have analyzed the statistics of differences in m⁵C distribution in the CDS region between different groups through post-hoc tests for two-by-two comparisons were performed with adjustment using the Benjamini & Hochberg method. The results showed that significant differences in m⁵C distribution in the CDS region were observed in downregulated or upregulated transcripts compared with unchanged transcripts upon ALYREF knockdown (Fig. 3F in the revised manuscript). However, downregulated transcripts but not upregulated transcripts bound by ALYREF significantly tended to undergo m⁵C modification in CDS regions (Fig. 3G in the revised manuscript). The corresponding significant difference was shown in the figures in the revised manuscript.

2. Fig. 3I-K. The color legends are unclear and need to be improved.

Our reply: We appreciate the reviewer's valuable comments. We have improved the color legends of Fig. 3I-K in the revised manuscript.

3. Lines 149-151. "Western blotting verified that PARP10 protein was significantly decreased in ovarian cancer cells following ALYREF depletion (Fig. 4a)." The Western blot results were not quantified and need to be to support this statement.

Our reply: Thanks for the reviewer's valuable comments. We have added grayscale statistics results for all western blots to support the statement in the revised manuscript.

4. Discussion. Much of the Discussion repeats results already presented in prior sections of the manuscript and should be revised to focus more on evaluation of results.

Our reply: As suggested, we have revised the discussion section in the revised manuscript.

Reviewer #3 (Remarks to the Author):

The presented research article entitle “ALYREF forms nuclear phase condensates and stabilizes m⁵C-modified PARP10 mRNA to promote ovarian cancer progression”, provides original and novel insights into the role of the 5-methylcytosine (m⁵C) RNA modification in the tumorigenesis of various cancers, particularly ovarian cancer. The study focuses on ALYREF, a m⁵C recognition protein whose role and underlying mechanism in ovarian cancer had not been previously elucidated. The researchers demonstrate that ALYREF forms condensates in the nucleus through a process called liquid-liquid phase separation. They further establish that these condensates contribute to ovarian cancer malignancy by promoting the proliferation and metastasis of ovarian cancer cells. These findings suggest that ALYREF's phase separation process facilitates the malignant progression of ovarian cancer by upregulating PARP10 expression in an m⁵C-dependent manner.

Overall, while **the article presents some interesting findings regarding the role of ALYREF in forming liquid-liquid phase (LLPS) condensates**, it lacks sufficient originality and significance. It has been previously established that ALYREF functions as a "reader" of the 5-methylcytosine (m⁵C) RNA modification and regulates the nuclear export of RNA to the cytoplasm.

Although the authors propose an intriguing mechanism involving the formation of LLP condensates by ALYREF, there is a notable gap in understanding the precise mechanism that connects m⁵C-modified mRNAs, condensate formation, nuclear export, and the degradation of modified mRNAs. This missing link hinders the overall novelty and significance of the article, as it fails to provide a comprehensive understanding of the functional implications and regulatory mechanisms underlying ALYREF-mediated processes in the context of m⁵C-modified mRNAs. Further

exploration and elucidation of this missing mechanistic aspect would greatly enhance the originality and significance of the research findings. In addition, for a better immersion into the study, some other parts of the work need to be clarified to avoid premature conclusions.

Our reply: We are grateful for the reviewer's acknowledgment of our work. However, we disagree with the reviewer's comments on the originality and significance of our manuscript. In this study, we first showed that ALYREF forms condensates in the nucleus by liquid-liquid phase separation and promotes ovarian cancer malignancy by increasing the proliferation and metastasis of ovarian cancer cells. ALYREF binds to m⁵C-modified PARP10 mRNA competitively with MTR4, thereby enhancing the stability and nuclear export of PARP10 mRNA and activating downstream PI3K-AKT signaling. LLPS is indispensable for ALYREF promoting PARP10 expression and mRNA stability as well as ovarian tumorigenesis. Furthermore, ALYREF is highly expressed in cancers including ovarian cancer, and predicts a poor prognosis for cancer patients. Thus, targeting ALYREF phase separation might be promising for cancer treatment.

To strengthen the mechanisms underlying ALYREF condensation regulating m⁵C-modified PARP10 expression, we have performed a series of experiments. The straightforward mechanism is that ALYREF binding m⁵C-modified mRNAs promote their condensation, as m⁵C-modified RNA enhances ALYREF condensation and RNA FISH assays showed that PARP10 mRNA was localized in ALYREF condensates in the nucleus of ovarian cancer cells (Fig. EV5H in the revised manuscript). ALYREF condensation with PARP10 mRNA protected from MTR4-mediated RNA degradation and promoted mRNA stability and nuclear export as well as its expression. We constructed stable ovarian cancer cells overexpressed with ALYREF labeled with tGFP and conducted FRAP *in vivo* to confirm ALYREF LLPS in ovarian cancer cells (Fig. 5I, J, and Fig. EV5E, F in the revised manuscript). To map the key domains for its phase separation, we constructed the mutants of ALYREF. *In vitro* phase separation assays demonstrated that the N-terminal IDR and

C-terminal IDR together drove LLPS of ALYREF, and ALYREF phase separation disappeared when the IDRs were absent. Moreover, mutations of Arg and Lys in IDRs to Ala also decreased ALYREF condensation (Fig. 6B in the revised manuscript). The aggregates were significantly smaller, and PARP10 upregulation was disrupted in cells expressing the ALYREF mutants compared with cells with wild-type ALYREF expression (Fig. 6C, D in the revised manuscript). Also, the opto-droplet system was constructed, and photo induced IDR phase separation assays were carried out. The results show that the phase separation phenomenon of ALYREF occurs in the case of ALYREF full-length (Fig. 6E, F in the revised manuscript).

To confirm that LLPS was involved in ALYREF regulation of PARP10 expression, we constructed GFP-ALYREF and its respective truncated forms in which IDRs were depleted or mutated and transfected the constructs into ovarian cancer cells. RT-qPCR showed that unlike wild-type ALYREF, which promoted PARP10 expression, all the mutated forms of ALYREF exerted little effect on the level of PARP10 mRNA (Fig. 6G and Fig. EV6A in the revised manuscript). The nucleoplasmic isolation and RNA stability assays also showed that the mutated forms of ALYREF did not affect the nucleoplasmic distribution and stability of PARP10 mRNA (Fig. 6H, I, and Fig. EV6B in the revised manuscript). Immunoprecipitation assays revealed that only a mutation of R/K to A destroyed the interaction of ALYREF with EIF4A3 and MAGOH, whereas other LLPS-depleted ALYREF mutants were intactly bound to EIF4A3 and MAGOH. ALYREF interacts with other proteins primarily through amino acid sequences in the 55-182 region^[5]. These results suggest that LLPS depletion interferes with ALYREF-regulating PARP10 expression and is independent of its interaction with other components in the TREX (Fig. EV6C in the revised manuscript).

ALYREF overexpression in ovarian cancer (OC) should be validated by either Western blotting (WB) or immunohistochemistry (IHC) using OC samples.

PARP10 overexpression in OC should be validated by either WB or IHC using OC

samples, and its correlation with ALYREF should be examined.

Our reply: Thanks for the reviewer's valuable comments. We have conducted western blotting to validate ALYREF and PARP10 expression in OC samples. ALYREF and PARP10 were highly expressed in ovarian cancer tissues (Fig. 7K in the revised manuscript). ALYREF expression was positively correlated with PARP10 expression in ovarian cancer (Fig. 7L in the revised manuscript).

The authors need to demonstrate that the ALYREF mutant does not bind to m⁵C-methylated mRNAs.

Our reply: We sincerely appreciate the reviewer's valuable comments. Fig. 4L in the revised manuscript showed that the ALYREF mutant did not bind to m⁵C-methylated PARP10 mRNA when the K171 site of ALYREF was mutated. The K171 site is a key site for ALYREF to recognize m⁵C-modified RNA^[6]. The binding capacity of ALYREF to m⁵C-modified oligonucleotides was significantly higher than that of the control group without m⁵C methylation.

Given the low prevalence of m⁵C sites in mRNAs, it is essential to address the discrepancy between the reported number of potential m⁵C modification sites and the expected number. The bisulfite sequencing library preparation and bioinformatic analysis should be repeated with more stringent criteria. For instance, the authors should consider using higher temperature or longer bisulfite treatments to ensure accurate RNA deamination and avoid false positives. It is recommended to demonstrate deamination in 95% of all aligned sequences.

Our reply: We sincerely appreciate the reviewer's valuable comments. We performed a quality control analysis on the original sequencing data. It is generally believed that after bisulfite treatment, the mutation rate of base C is more than 99%, indicating that the method of bisulfite treatment is very good^[7]. The following figure is the quality control diagram we made based on the original sequencing data.

The content level of the four bases in the input group is basically the same (Figure left

panel). After treatment with bisulfite, the base C without m⁵C modification was converted to T. In the diagram, we can see a significant increase in base T content. The content of base C modified by m⁵C was less than 1%. This result demonstrates that our m⁵C processing process is qualified (Figure right panel shown below).

In Figure 4I, the authors have shown that the ALYREF mutant fails to bind PARP10 mRNA. It is suggested to perform a RNA immunoprecipitation (RIP) assay using a mutant form of PARP10 where the methylated C cannot be methylated.

Our reply: We sincerely appreciate the reviewer's valuable comments. We have conducted an ALYREF RIP assay using a mutant form of PARP10 where the methylated C cannot be methylated. Compared with wild-type PARP10 mRNA, mutation of the m⁵C site on PARP10 mRNA significantly abrogated its interaction with ALYREF (Fig. 4S in the revised manuscript).

To further validate the methylation of PARP10, the authors should identify the candidate methylase responsible for this modification.

Our reply: Thanks for the comments. We will identify the candidate methylase responsible for the m⁵C modification of PARP10 mRNA. Especially, it has been reported that NOP2/Sun RNA methyltransferase family member 2 (NSUN2) is the main methyltransferase in mRNA m⁵C modification^[6]. To further validate the methylation of PARP10, RIP assays showed that NSUN2 bound PARP10 mRNA in ovarian cancer cells (Fig. 4I and Fig. EV4H in the revised manuscript). The meRIP

assay showed that m⁵C modification of PARP10 mRNA was reduced when NSUN2 was knocked down in ovarian cancer cells (Fig. 4J in the revised manuscript). RIP assay results also indicated that the binding ability of ALYREF to PARP10 mRNA was reduced when NSUN2 was knocked down (Fig. 4K and Fig. 4I in the revised manuscript). In addition, NSUN2 knockdown decreased PARP10 expression in ovarian cancer cells (Fig. 4J, K in the revised manuscript). These results suggest NSUN2 mediates m⁵C modification of PARP10 mRNA in ovarian cancer cells.

The authors need to demonstrate that the mRNA levels of the mutant form of PARP10, where C cannot be methylated, are not regulated by ALYREF expression and are degraded regardless of ALYREF or MTR4 expression. Additionally, the authors should observe similar effects when the methylase that methylate PARP10 mRNA is silenced.

Our reply: We highly appreciate the comments. As suggested, we constructed an expression vector of PARP10 with mutations in the m⁵C site. The following meRIP assay illustrated that m⁵C modification of PARP10 mRNA was decreased upon mutation of the m⁵C site on PARP10 mRNA (Fig. 4R in the revised manuscript). Consistently, mutation of the m⁵C site on PARP10 mRNA significantly abrogated its interaction with ALYREF (Fig. 4S in the revised manuscript). Moreover, knockdown of ALYREF or NSUN2 decreased the expression of wild-type PARP10 mRNA but not the m⁵C site-mutated PARP10 mRNA, whereas knockdown of MTR4 increased the expression of the m⁵C site-mutated PARP10 mRNA but not wild-type PARP10 mRNA (Fig. 4T-V in the revised manuscript). RNA half-life assays showed that knockdown of ALYREF or NSUN2 decreased the stability of wild-type PARP10 mRNA but not the m⁵C site-mutated PARP10 mRNA (Fig. EV4M in the revised manuscript). These results suggest that ALYREF promotes PARP10 expression by regulating the nuclear export and stability of PARP10 mRNA in an m⁵C-dependent manner.

The authors have shown that methylated PARP10 mRNA is blocked in the nucleus in

ALYREF knockdown cells, leading to PARP10 degradation. As ALYREF is also responsible for the formation of nuclear LLPS droplets, the authors should demonstrate that PARP mRNA is also condensed within these droplets.

Our reply: Insightful comments! Figure 6E showed that ALYREF is also responsible for the formation of nuclear LLPs droplets, and PARP10 upregulation is disrupted in the ALYREF mutants compared with cells with wild-type ALYREF. We have performed RNA fluorescence in situ hybridization assays to observe the localization of PARP10 RNA in the condensates of ALYREF, and results showed that PARP10 mRNA was localized in ALYREF condensates in the nucleus of ovarian cancer cells (Fig. EV5H in the revised manuscript).

The conclusions drawn regarding the role of ALYREF as a reader of m⁵C modifications and its connection to the formation of RNA-protein condensates and subsequent export to the cytoplasm lack robustness, validity, and reliability.

While the study provides some intriguing insights into these processes, the evidence presented is insufficient to establish a solid foundation for these mechanisms. The experimental data and analyses provided do not sufficiently support the proposed mechanisms and their direct relationship to ALYREF's function as an m⁵C-reader. The study would benefit from additional experiments, rigorous validation, and more extensive data analysis to strengthen the reliability and validity of the conclusions. It is crucial to ensure that the findings are supported by robust evidence, reproducible experiments, and comprehensive statistical analyses to enhance the scientific rigor and credibility of the study's conclusions.

Our reply: We are grateful for the reviewer's acknowledgement of our work. As suggested, we have performed a series of additional experiments to enhance the conclusions of ALYREF phase separation as well as its links with PARP10 regulation, as mentioned above.

Besides *in vitro* experiments, we constructed stable ovarian cancer cells overexpressed with ALYREF labeled with tGFP and conducted FRAP *in vivo*. The

results showed that the recovery times of the OVCAR3 cell line and the A2780 cell line were about 105 s and 66 s, respectively (Fig. 5I, J, and Fig. EV5E, F in the revised manuscript). RNA fluorescence in situ hybridization assay showed that PARP10 mRNA was localized in ALYREF condensates in the nucleus of ovarian cancer cells (Fig. EV5H in the revised manuscript). To map the key domains for its phase separation, we constructed the mutants of ALYREF. *In vitro* phase separation assays demonstrated that the N-terminal IDR and C-terminal IDR together drove LLPS of ALYREF, and ALYREF phase separation disappeared when the IDRs were absent. Moreover, mutations of Arg and Lys in IDRs to Ala also decreased ALYREF condensation (Fig. 6B in the revised manuscript). The aggregates were significantly smaller, and PARP10 upregulation was disrupted in cells expressing the ALYREF mutants compared with cells with wild-type ALYREF expression (Fig. 6C, D in the revised manuscript). To study the regions where ALYREF performs phase separation, the opto-droplet system was constructed, and photo induced IDR phase separation assays were carried out. The results show that the phase separation phenomenon of ALYREF occurs in the case of ALYREF full-length (Fig. 6E, F in the revised manuscript).

To examine whether LLPS was involved in ALYREF regulation of PARP10 expression, we constructed GFP-ALYREF and its respective truncated forms in which IDRs were depleted or mutated and transfected the constructs into ovarian cancer cells. RT-qPCR showed that unlike wild-type ALYREF, which promoted PARP10 expression, all the mutated forms of ALYREF exerted little effect on the level of PARP10 mRNA (Fig. 6G and Fig. EV6A in the revised manuscript). The nucleoplasmic isolation and RNA stability assays also showed that the mutated forms of ALYREF did not affect the nucleoplasmic distribution and stability of PARP10 mRNA (Fig. 6H, I, and Fig. EV6B in the revised manuscript). To detect whether ALYREF mutants with LLPS loss affected its interaction with other components in the TREX complex, we performed immunoprecipitation assays and found that only mutation of R/K to A destructed the interaction of ALYREF with EIF4A3 and

MAGOH, whereas other LLPS-depleted ALYREF mutants were intactly bound to EIF4A3 and MAGOH. ALYREF interacts with other proteins primarily through amino acid sequences in the 55-182 region^[5]. These results suggest that LLPS depletion interferes with ALYREF-regulating PARP10 expression and is independent of its interaction with other components in the TREX (Fig. EV6C in the revised manuscript).

Additionally, to confirm the effect of LLPS on ALYREF promoting ovarian cancer, truncations, mutants, and ALY-wt were overexpressed in ovarian cancer cells, respectively. ALY-wt overexpression promoted the proliferation, migration, and invasion abilities of ovarian cancer cells (Fig. 7A-C in the revised manuscript). However, LLPS depletion of ALYREF failed to facilitate ovarian cancer cell proliferation, migration, and metastasis (Fig. 7A-C in the revised manuscript). Consistently, *in vivo* assays also showed that LLPS depletion abrogated the promoting effect of ALYREF on tumor growth in mice (Fig. 7D-F in the revised manuscript). Together, these results demonstrate that ALYREF promotes PARP10 expression and tumorigenesis dependently on LLPS.

The article lacks a comprehensive description of the statistical analyses employed and the treatment of uncertainties.

Our reply: According to the suggestion of the reviewer, we will add a comprehensive description in which we employed statistical analyses and the treatment of uncertainties.

Minor comments:

In Figure 2c, the chosen color for each condition changes in each panel, which can be confusing. It is recommended to use consistent colors for clarity and ease of interpretation.

Figure 8 does not contribute significant information to the article and could be incorporated into Figure 1 for better presentation and organization.

Our reply: As required, we have revised the manuscript. Consistent colors were used in Figure 2C in the revised manuscript. In Figure 8 and Figure EV8 in the revised manuscript, we reanalyzed the clinical significance of ALYREF in ovarian cancer including its subtypes as well as other cancers, and provided new results. Especially, ALYREF was found to be amplified and upregulated in various cancers and predict poor prognosis, which demonstrates the significant implication of ALYREF in the pan-cancer. Alternatively, we would also revise this as suggested.

Reference

1. Khatib JB, Dhoonmoon A, Moldovan GL, Nicolae CM. PARP10 promotes the repair of nascent strand DNA gaps through RAD18 mediated translesion synthesis. *Nat Communication*. 2024, 15(1):6197.
2. Khatib JB, Schleicher EM, Jackson LM, Dhoonmoon A, Moldovan GL, Nicolae CM. Complementary CRISPR genome-wide genetic screens in PARP10-knockout and overexpressing cells identify synthetic interactions for PARP10-mediated cellular survival. *Oncotarget*. 2022, 13:1078-1091.
3. Tian L, Yao K, Liu K, Han B, Dong H, Zhao W, Jiang W, Qiu F, Qu L, Wu Z, Zhou B, Zhong M, Zhao J, Qiu X, Zhong L, Guo X, Shi T, Hong X, Lu S. PLK1/NF- κ B feedforward circuit antagonizes the mono-ADP-ribosyltransferase activity of PARP10 and facilitates HCC progression. *Oncogene*. 2020, 39(15):3145-3162.
4. Zhou Z, Wei B, Liu Y, Liu T, Zeng S, Gan J, Qi G. Depletion of PARP10 inhibits the growth and metastatic potential of oral squamous cell carcinoma. *Front Genet*. 2022, 13:1035638.
5. Pacheco-Fiallos B, Vorländer MK, Riabov-Bassat D, Fin L, O'Reilly FJ, Ayala FI, Schellhaas U, Rappsilber J, Plaschka C. mRNA recognition and packaging by the human transcription-export complex. *Nature*. 2023, 616(7958):828-835.
6. Yang X, Yang Y, Sun et al. 5-methylcytosine promotes mRNA export-NSUN2 as the methyltransferase and ALYREF as an m5C reader. *Cell Res*. 2017 May;27(5):606-625.
7. Rieder D, Finotello F. Analysis of high-throughput RNA bisulfite sequencing data. *RNA Methylation: Methods and Protocols*, 2017: 143-154.

Dear Dr Yi,

Thank you again for transferring your amended manuscript (EMBOJ-2025-120620-T) to The EMBO Journal, together with a point-by-point response to the issues raised during peer-review at the previous venue. Please accept my sincere apologies for the unusual protraction with the reassessment due to delayed expert input and detailed discussion in the editorial team. Your revised study was sent back to the three referees for their scientific reassessment, and we have received detailed re-reports from one of them, which I enclose below. Please note that while referees #2 and #3 were at this time not able to look back into your study, we have editorially considered your response to their concerns and found them to be addressed satisfactorily. As

you will see, referee #1 states that the work has been substantially enhanced by the revisions and s/he is now in favour of publication, pending minor revision.

Thus, we are pleased to inform you that your manuscript has been accepted in principle for publication in The EMBO Journal.

Please carefully consider the remaining minor points raised by referee #1 by adding complementary data and-or complementing the manuscript text where appropriate.

Also, we now need you to take care of a number of minor issues related to formatting and data annotation, which I will share shortly in a separate message, together with additional changes and requests by our production team for Source Data provision.

Please submit a revised version of the manuscript using the link enclosed below, addressing the advisor's comments.

As you might have seen on our web page, every paper at the EMBO Journal now includes a 'Synopsis', displayed on the html and freely accessible to all readers. The synopsis includes a 'model' figure as well as 2-5 one-short-sentence bullet points that summarize the article. I would appreciate if you could provide this figure and the bullet points.

Thank you again for giving us the chance to consider your manuscript for The EMBO Journal, I look forward to hearing from you and receiving your final revised version of the manuscript.

Kind regards,

Daniel Klimmeck

Please use the link below to submit your revision:

Referee #1:

The authors have strongly modified the manuscript based on the comments of the reviewers. Mechanistically, they have now more experimental evidence supporting their model.

Major comments:

1. The prognostic impact of ALYREF in ovarian cancer is in my opinion less well worked out. Did the authors perform multivariate analyses? It seems that stage and grade are very important factors. Subanalyses in stage 3 or stage 4 patients would help. What are the characteristics of the patient group, for instance chemotherapy, stage, and grade? These should be included in a table. If ALYREF and PARP10 mRNA expression are closely related, is PARP10 expression then related to PFS and overall survival?
2. Has overexpression of the various ALYREF mutant (Fig7A-F) any effect on PARP10 protein expression?
3. In figure 7G and H P10 is used, but the abbreviation is not explained.

Dear Dr Yi,

Thank you for your patience. Further top below, I enclose the last of additional formatting requirements for your article.

Please let us know any time should you have questions related.

We look forward to your final resubmission.

Kind regards,

Daniel Klimmeck

>>Authors: Please add co-corresponding author T.L.'s institutional email address on the manuscript title page and in our system.

>> Author Contributions: Remove the author contributions information from the manuscript text. Note that CRediT has replaced the traditional author contributions section as of now because it offers a systematic machine-readable author contributions format that allows for more effective research assessment. and use the free text boxes beneath each contributing author's name to add specific details on the author's contribution.

More information is available in our guide to authors.
<https://www.embopress.org/page/journal/14602075/authorguide>

>> Provide a completed Author Checklist.

>> Section order should be corrected as follows: title page with complete author information, abstract, keywords, introduction, results, discussion, methods, data availability section, acknowledgements, disclosure and competing interests statement, references, main figure legends, tables, expanded figure legends.

Add the 'Statistics' paragraph to the Methods section.

Integrate the 'Study approval' information into the 'In vivo tumorigenesis and metastasis assays' section.

>> Figures in separate files: Please upload the EV figures as individual, high resolution figure files, and add the EV figure legends to the manuscript text, under the heading 'Expanded View Figure Legends', placed after the main figure legends.

>> Figure callouts: please correct the callouts for Tables EV1-EV7 to Dataset EV1 - EV7, a well as the callouts for Suppl. Table S7. Please also correct the callouts for Movies EV1 and EV2.

>> References: please adjust reference format to EMBO Journal format, 10 authors et al. . dois should be removed

>> Funding: please enter all funding information into the list of funders in our online system and detail them in addition in the Acknowledgments section of the manuscript. 'The 67th batch China Postdoctoral Science Foundation2020M673560XB, Chongqing Talent Program (cstc2024ycjh-bgzxm0077), Chongqing Graduate Mentor Team Construction Project from Chongqing Education Commission, Program for Youth Innovation in Future Medicine, and the Chongqing Medical University (W0058)' are currently missing in our system.

>> Dataset EV legends: The seven tables should be renamed "Dataset EV1" - EV7. Each file needs a legend added in a separate tab/worksheet. The two movies play; please add a legend for each movie.

>> Please add a Reagents and Tools table to the Methods section, as a separate file using the existing template in the Guide For Authors, listing key reagents, experimental models, software and relevant equipment.

>> Please provide source data for the study as to the separate request e-mail. Source data should be uploaded as one (zipped) file per figure.

>> Data availability section: Please provide URLs for the sequencing datasets and ensure privacy is released and the data sets are public.

>> Avoid textual redundancy in the introduction, results and discussion sections with your earlier 2023 study (PMID 37639158).

>> Consider additional changes and comments from our production team as indicated below:

DATA CHECK: FAIL

- Data Availability section:

1. Please note that the specific URLs for PRJNA830530, PRJNA830297, PRJNA830639 datasets are not provided in the data availability statement.

- Figure legends:

1. Please define the annotated p values ****/***/**/* as well as provide the exact p-values for the same in the legend of figure EV2 A; EV4 B, F, H, I, K; EV7 B as appropriate.

2. Please note that the exact p values are not provided in the legends of figures 1A, B, C, D, E, F, G, H, I, K, L, N; 2C, E, G; 3I-N; 4B, D, F, G, I, J, K, M, N, O, Q, R, S, T, U, V; 6G, H; 7A, B, C, E, F, G, H, I, K, L; 8D, E; EV1 A, C, E, I; EV5 G; EV8 C, G, H

3. Please indicate the statistical test used for data analysis in the legends of figures 1A, B, C, D, E, F, G, H, I, K, L, N; 2C, E, G, M; 3B, E, F, G, I-N; 4B, D, F, G, I, J, K, M, N, O, Q, R, S, T, U, V; 6G, H; 7A, B, C, E, F, G, H, I, J, K, L; 8D, E; EV1 A, C, E, I; EV2 A, B; EV3 B; EV4 B, C, D, F, H, I, K; EV4 B, F, H, I, K; EV5 G, EV7 B; EV8 C, G, H

4. Please note that the box plots need to be defined in terms of minima, maxima, centre, bounds of box and whiskers, and percentile in the legends of figures 1B, C; 3E, 8C, EV8 C, G, H

5. Please note that information related to n is missing in the legends of figures 1A, B, C, D, E, F, G, H, I, K, L, N; 2C, E, G; 3E, I-N; 4B, D, E, F, G, I, J, K, M, N, O, Q, R, S, T, U, V; 5H, J; 6D, G, H, I; 7A, B, C, E, F, G, H, I, K; 8C, D, E; EV1 A, C, E, I; EV4 B, C, D, F, H, I, K, M; EV5 F, G; EV7 B; EV8 C, G, H

6. Please note that the error bars are not defined in the legends of figures 5H, J; 8D, E; EV1 A, C, E, I; EV4 B, C, D, F, H, I, K, M; EV5 F, G

7. Please note that scale bar and its definition are missing for figures 5A, 6B, F; EV5 H, EV7 B.

8. Please note that the red arrows are not defined in the legend of figure 5G, I; EV5 E. This needs to be rectified.

Point-by-Point Responses to the Comments from Reviewers

The authors have strongly modified the manuscript based on the comments of the reviewers. Mechanistically, they have now more experimental evidence supporting their model.

Our reply: We are grateful for the reviewer's acknowledgment of our work and constructive comments.

Major comments:

1. The prognostic impact of ALYREF in ovarian cancer is in my opinion less well worked out. Did the authors perform multivariate analyses? It seems that stage and grade are very important factors. Subanalyses in stage 3 or stage 4 patients would help. What are the characteristics of the patient group, for instance chemotherapy, stage, and grade? These should be included in a table. If ALYREF and PARP10 mRNA expression are closely related, is PARP10 expression then related to PFS and overall survival?

Our reply: We really appreciate the reviewer's comments. As suggested, we analyzed the ALYREF expression in ovarian cancer through a tissue microarray including 134 ovarian cancer patients and found that ALYREF expression was significantly associated with poor survivals (Fig. EV8G,H in the revised manuscript). Multivariate analyses revealed that ALYREF high expression predicted the poor progression-free survival, but no significance between ALYREF expression with patients' overall survival (Fig.1 shown below), which might be attribute to the limited patient number in our cohort. Thus, we didn't include this data into the revised manuscript. Instead, we have included a table of patents with clinicopathological characteristics (Dataset EV7 in the revised manuscript). Moreover, we have performed subanalyses in advanced stages (stage III or stage IV) patients. Higher expression of ALYREF portended a worse prognosis for ovarian cancer patients, regardless of p53 mutation status or tumor grades (Fig. EV8E,F). However, higher ALYREF expression was correlated with shorter survivals in patients at advanced stages (III and IV) (Fig.

EV8H in the revised manuscript). Furthermore, we analyzed the associations of PARP10 expression with patients' survivals and found that higher expression of PARP10 predicted worse prognosis for ovarian cancer patients (Fig. EV8D in the revised manuscript).

Figure 1. Multivariate analyses show the associations of ALYREF expression with patients' survivals.

2. Has overexpression of the various ALYREF mutant (Fig7A-F) any effect on PARP10 protein expression?

Our reply: We sincerely appreciate the reviewer's comments. As suggested, we have detected PARP10 protein expression in ovarian cells with overexpression of the various ALYREF mutants. Consistently, overexpression of wild-type ALYREF significantly promoted PARP10 protein expression, however, its mutants could decrease this effect (Fig. EV6A in the revised manuscript).

3. In figure 7G and H P10 is used, but the abbreviation is not explained.

Our reply: We apologize for the oversight and have revised the manuscript with clear descriptions.

Dear Dr Yi,

Thank you for submitting the revised version of your manuscript. I have now evaluated your amended manuscript and concluded that the remaining minor concerns have been sufficiently addressed.

I am thus pleased to inform you that your manuscript has been accepted for publication in the EMBO Journal.

Kind regards,

Daniel Klimmeck

Daniel Klimmeck, PhD
Senior Editor
The EMBO Journal
EMBO
Postfach 1022-40
Meyerhofstrasse 1
D-69117 Heidelberg
contact@embojournal.org

Please note that it is The EMBO Journal policy for the transcript of the editorial process (containing referee reports and your response letters) to be published as an online supplement to each paper. If you should prefer removal of any referee-only figures included in the point-by-point response(s), e.g. because they may still be used for future publication or because they have been reproduced from published work by others, please do let us know immediately via response email.

More information is available here: https://www.embopress.org/transparent-process#Review_Process